# Lipid-polymer nanoparticles to probe the native-like environment of intramembrane rhomboid protease GlpG and its activity

Henry Sawczyc [1], Takashi Tatsuta [2], Carl Öster[1], Spyridon Kosteletos [1], Sascha Lange[1], Claudia Bohg [1], Thomas Langer [2] & Adam Lange [1,3] ✉

Polymers can facilitate detergent-free extraction of membrane proteins into nanodiscs (e.g., SMALPs, DIBMALPs), incorporating both integral membrane proteins as well as co-extracted native membrane lipids. Lipid-only SMALPs and DIBMALPs have been shown to possess a unique property; the ability to exchange lipids through 'collisional lipid mixing'. Here we expand upon this mixing to include protein-containing DIBMALPs, using the rhomboid protease GlpG. Through lipidomic analysis before and after incubation with DMPC or POPC DIBMALPs, we show that lipids are rapidly exchanged between protein and lipid-only DIBMALPs, and can be used to identify bound or associated lipids through 'washing-in' exogenous lipids. Additionally, through the requirement of rhomboid proteases to cleave intramembrane substrates, we show that this mixing can be performed for two protein-containing DIBMALP populations, assessing the native function of intramembrane proteolysis and demonstrating that this mixing has no deleterious effects on protein stability or structure.

The study of membrane protein function, particularly the study of lipid-protein interactions, remains difficult. Most notably, the use of detergent in protein extraction and purification leads to a loss of the native lipid environment. Whilst membrane proteins are often reconstituted into a model membrane system for structural and functional characterization, the delipidation by detergents can destroy any weak protein-lipid interactions that could be vital for native function and stability of the protein.

In recent years, there has been a renewed effort to study membrane systems in their native environment[1], and the importance of understanding lipid-protein interactions is becoming increasingly recognized[2,3]. Lipid-polymer nanoparticles (otherwise known in the literature as 'nanodiscs' and referred to in this work as such with the caveat that the particles formed from polymer addition have been observed to be a mixture of disc-shaped and other globular particles[4]). SMALPs and DIBMALPs, have been shown to extract integral membrane proteins along with their native lipids into disc-shaped particles[5,6], without the use of detergents at any step. Proteins within these disks often possess increased stability compared to their detergent-solubilized counterparts, partly due to the retention of the native-like lipid environment and the increased lateral pressure[7,8]. Due to this co-extraction of native lipids, and the relatively low ionization energy required over equivalent detergent samples, SMALPs/DIBMALPs have proven to be a revolutionary tool in the rapidly developing field of 'native-omics'[9]. Similarly, peptide-based nanodiscs have shown some propensity for lipid mixing, either through single lipid diffusion[10], or through nanodisc fusion[11–13]. However, they are still constrained by the dependence on formation methods that involve detergents.

Rhomboid proteases are intramembrane serine proteases, cleaving their integral membrane substrates within the lipid bilayer[14]. An increasing number of rhomboid proteases, including the *E. coli* model rhomboid protease GlpG, are known to locally thin the lipid membrane

[1]Research Unit Molecular Biophysics, Leibniz-Forschungsinstitut für Molekulare Pharmakologie, Robert-Rössle-Straße 10, 13125 Berlin, Germany. [2]Max-Planck-Institute for Biology of Ageing, Department of Mitochondrial Proteostasis, Joseph-Stelzmann-Str. 9b, 50931 Cologne, Germany. [3]Institut für Biologie, Humboldt-Universität zu Berlin, Invalidenstraße 42, 10115 Berlin, Germany. ✉e-mail: alange@fmp-berlin.de

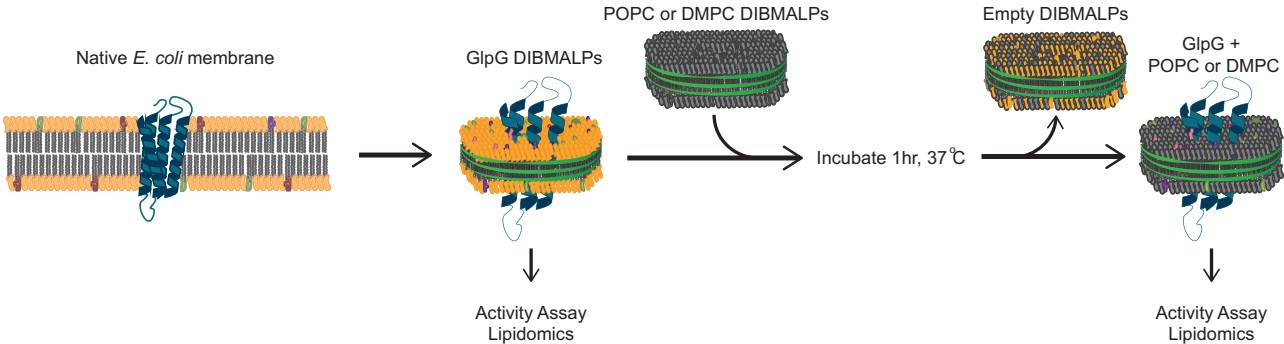

**Fig. 1 | Diagrammatic view of methods used in this work.** Schematic of the method utilized in this work, where natively extracted GlpG or GlpGΔN in DIB-MALPs (represented here as nanodiscs for visual ease) are incubated with POPC (1-palmitoyl-2-oleoyl-glycero-3-phosphocholine) or DMPC (1,2-dimyristoyl-sn-glycero-3-phosphocholine) DIBMALPs to 'wash-in' non-native neutral lipids, followed by removal by Ni-NTA purification. Activity and lipidomic analysis samples were taken to compare the effect of the native lipid environment vs POPC or DMPC in the washed DIBMALPs.

around the protein. This has been theorized to aid the diffusion of catalytically active rhomboid proteases through the membrane in search of substrates[15]. Additionally, this lipid remodeling or thinning suggests a potential function for catalytically inactive rhomboid proteases, known as iRhoms[16]. A recent study by Engberg et al. has shown that there is a specific requirement for phosphatidylethanolamine (PE) head group lipids for this lipid thinning[17].

GlpG has previously been solubilized by both SMA (styrene-maleic acid) and DIBMA (diisobutylene-maleic acid), showing more native-like activity with fluorophosphonate probe labeling (TAMRA-FP) and enhanced stability than detergent-solubilized GlpG[18,19]. It has also been shown that the cytosolic N-terminal domain may be responsible for the correct positioning of GlpG in the membrane[15], and it may directly interact with membrane lipids[18]. As cleavage by rhomboid proteases requires the substrate and protease to be in the same lipid bilayer, it could be suggested that the potential use of polymer technology is limited for this class of proteases. Recently, however, SMALP/DIB-MALPs have been shown to transfer lipids from one nanoparticle to another in solution[20]. This property, coined as 'collisional lipid mixing' by Keller et al., has, to date, only been shown between lipid-only SMALPs and DIBMALPs[20] and between a protein-containing SMALP and deuterated lipid-only SMALP[21]. A similar, albeit slower, technique of nanodiscs fusion has been used recently in peptide-based nanodiscs using monomer diffusion to observe the preferential lipid association of the *E. coli* ammonium transporter, AmtB[22] and Aquaporin Z[10].

The work presented in this study expands upon the previously reported mixing to show that it also occurs in protein-containing DIBMALPs. Through a combination of lipidomics and observation of GlpG protease activity pre- and post-lipid exchange, we show that 'collisional lipid mixing' put forward by Keller *et al.* is not limited to lipid-only polymer nanodiscs. This phenomenon can be used for the observation of weakly-interacting protein-lipid interactions, and to measure the activity of intramembrane protein activity, such as rhomboid proteases, as demonstrated in the workflow in Fig. 1.

## Results

### Lipid composition of *E. coli* membrane vs DM- and DIBMA-solubilized GlpG

The composition and relative abundances of the lipids co-purified with DM and DIBMA were analyzed for both GlpG (Fig. 2C, D) and GlpGΔN (Fig. 2G, H). The total phospholipid composition of the membrane is available in Supplementary Table 1. Due to GlpG and GlpGΔN being unable to be produced in the same cell line of *E. coli*, controls of whole-cell extract of BL21(C41) for GlpG (Fig. 2A, B) and BL21(C43) for GlpGΔN (Fig. 2E, F) were taken as a baseline for membrane lipid composition.

Interestingly, the addition of IPTG to BL21 C41 (DE3) cells transformed with the corresponding plasmid (Fig. 2A, B) causes a large

decrease in PE lipid composition (from 76.2% to 59.8%), and a subsequent increase in both PG (phosphatidylglycerol) and CL (cardiolipin) lipids (from 12.4% to 18.3% and 11.3% to 21.8%, respectively). This change in lipid composition upon IPTG induction has similarly been observed by Teo et al. with *E. coli* BL21(DE3) cells[23].

No significant differences in lipid headgroup composition can be observed after DM or DIBMA solubilization and GlpG purification (Fig. 2C, D) when compared to the induced BL21 C41 (DE3) sample, with the exception of a slight increase in PG lipid for the DIBMA-solubilised GlpG sample. This increased PG uptake by maleic acid containing polymers (of which DIBMA is a member) has been observed previously by Barniol-Xicota et al.[24].

No significant change between the uninduced and induced whole-cell extract of BL21(C43) is observed (Fig. 2E, F). However, upon DM or DIBMA solubilization and subsequent GlpGΔN purification, a notable decrease in PE lipids and increase in CL lipids can be observed when compared to the induced BL21 C43 (DE3) sample.

Overall, GlpG and GlpGΔN extracted into DIBMALPs have approximately 6× the lipid:protein ratio of GlpG and GlpGΔN extracted into DM micelles (Table S3). This is indicative of partial delipidation by the DM detergent during solublisation−where only tightly associated lipids are retained.

### Replacement of native membrane lipids in DIBMALPs with exogenous lipids

Exchange of natively extracted membrane lipids in GlpG-DIBMALPs or GlpGΔN-DIBMALPs (Fig. 2D, H, respectively) through the addition of 100× molar excess of either DMPC or POPC lead to an overall decrease in native lipids and an increase in either DMPC or POPC (respective to the lipid co-incubated). Overall, GlpG-DIBMALPs were more receptive to PC lipid incubation, with a replacement of native lipids to 61.3% (±12.4%) DMPC or 66.4% (±14.2%) POPC (Fig. 3A, B), whereas GlpGΔN retained more native lipids overall, with PC lipid incubation increasing exogenous lipid composition only to 45.7% (±10.4%) and 22.3% (±3.4%) for DMPC and POPC, respectively (Fig. 3C, D).

Significantly retained lipids, i.e., those which were in the top 5 highest relative concentrations in any PC-washed sample of GlpG or GlpGΔN are shown in Fig. 3E−O. All of these lipids are detected in high concentrations in either BL21 C41 (DE3) or BL21 C43 (DE3) whole-cell extracts (such as POPE and POPG in Fig. 3I, L, respectively). This indicates the absence of functionally relevant lipid-protein interactions.

### Assessment of protease activity in native-like and PC-rich environments

Activity of GlpG and GlpGΔN solubilized by DM was initially assessed through TAMRA-FP labeling (Supplementary Fig. 1D)[25], as well as cleavage of FLAG-TatA-SUMO (Supplementary Fig. 3D, E)[26], and

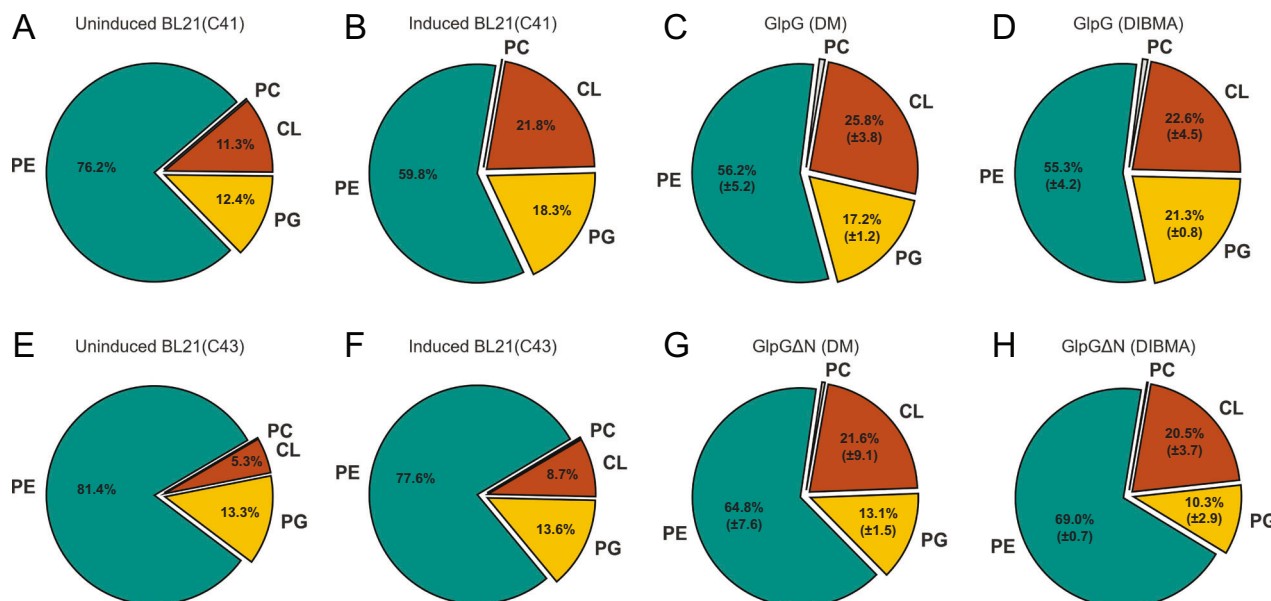

**Fig. 2 | Summary of extracted lipids for samples collected and analyzed by lipidomics.** Lipid head group composition as a percentage of total lipid for: **A** Whole-lipid extract of uninduced *E. coli* BL21 C41 (DE3) cells. **B** Whole-lipid extract of IPTG-induced *E. coli* BL21 C41(DE3) cells. **C** DM-solubilised and Ni-NTA purified GlpG, purified from *E. coli* BL21 C41(DE3). **D** DIBMA-solubilized and Ni-NTA purified GlpG, purified from *E. coli* BL21 C41(DE3). **E** Whole-lipid extract of uninduced *E. coli* BL21 C43(DE3) cells. **F** Whole-lipid extract of IPTG-induced *E. coli* BL21 C43(DE3) cells. **G** DM-solubilized and Ni-NTA purified GlpGΔN, purified from *E. coli* BL21 C43(DE3). **H** DIBMA-solubilized and Ni-NTA purified GlpGΔN, purified from *E. coli* BL21 C43(DE3). Errors given are standard deviation from three biological replicates. Full values are available in Supplementary Table 1.

through fluorescence measurement following the cleavage of TatA-FITC (Fig. 4A and Supplementary Fig. 4A–D)[27,28], all of which is broadly in line with previously published results for GlpG or GlpGΔN in *E. coli* membranes, albeit with slightly lower quality in regards to results pertaining to GlpGΔN for the FLAG-TatA-SUMO cleavage. The specificity of GlpG proteolytic activity was checked through MALDI-TOF experiments (Supplementary Fig. 5), where the products of TatA-FITC cleavage can be observed and the site of cleavage can be determined. However, due to the abundance of lipids and ionization difficulties, a clear profile of peptide fragments can only be observed for detergent-only samples (Supplementary Fig. 5A–F), as well as DIBMA samples (Supplementary Fig. 6), but not for LUV samples (Supplementary Fig. 5G–T). The peptide masses observed in all samples show the cleavage position of TatA-FITC by the protease, whereas the relative intensities are not quantitative to protease activity, due to the issues with ionization of the peptide from the membrane.

For DM-solubilized protease that is reconstituted in LUVs, there is a slightly higher activity observed in GlpGΔN than GlpG in *E. coli* lipids in the quantitative TatA-FITC cleavage, however, the main difference between the full-length and truncated GlpG is observed during reconstitution into non-native lipids (Fig. 4). It is worth noting that this difference was not observed in the cleavage of TatA-SUMO (Supplementary Fig. 3C), where no significant difference in activity between GlpGΔN in *E. coli* lipids and PC lipids can be observed in the gel-based assay.

For GlpG, reconstitution into POPC or DMPC membranes leads to an increase in activity compared to native-like *E. coli* membranes. This has similarly been shown by Engberg et al., albeit with a larger gain in activity observed[17]. For GlpGΔN, there was a significant decrease in proteolytic activity in DMPC or POPC liposomes. Both showed no significant difference in TAMRA-FP labeling, indicating that the activity difference observed is not due to a change in the structural integrity of the protease.

Solubilization and purification of both GlpG and GlpGΔN produced functionally active protease in DIBMALPs (Supplementary Fig. 3A–C), the purity of which was checked by SDS-PAGE

(Supplementary Fig. 1A, B). A comparison between 2D NCα solid-state NMR spectra of GlpG in DIBMALPs and GlpG in liposomes shows excellent agreement between the samples (Supplementary Fig. 2). This means that the overall structure of GlpG remains the same when extracted into DIBMALPs as when reconstituted into liposomes after purification in detergents, and, importantly, that no additional proteins are co-purified with GlpG during polymer extraction.

Measurement of substrate cleavage by GlpG and GlpGΔN in DIBMALPs, such as the similarly expressed and DIBMA-solubilised SUMO-TatA-FLAG (Supplementary Fig. 3A, C) and synthetically produced TatA-FITC (Fig. 4B and Supplementary Fig. 4E–J) is indicative of collisional mixing as proposed by Keller et al.[20] Outside of TAMRA-FP labeling (Supplementary Fig. 1B), the activity of GlpG and GlpGΔN measured in this work can only occur through mixing in bulk, due to the requirement for intramembrane interaction for proteolytic cleavage to occur. In addition, the detection of TatA-FITC cleavage fragments after incubation with GlpG and GlpGΔN DIBMALPs (Supplementary Fig. 6) shows that this cleavage matches the native activity for the protease.

In DIBMALPs, GlpG appears slightly more active than GlpGΔN in native-like lipid composition (Fig. 4B, "Initial"). Upon PC addition, GlpG experiences an increase in proteolytic activity, similar to that observed for DM-solubilised GlpG. However, a significant decrease in activity is observed for GlpGΔN after PC addition. Similar to the findings of Engberg et al., there is no corresponding change in TAMRA-FP labeling for GlpG nor GlpGΔN, indicating that the observed change in proteolytic activity is not due to a change in structural integrity. The activity data of biological replicates for both DM-solubilised and DIBMA-solubilised GlpG and GlpGΔN are shown separately due to the different levels of PC-enrichment between the samples.

Interestingly, analysis of TatA-FITC fragments after incubation with PC-washed GlpGΔN (Supplementary Fig. 6K, L, N, Q, R) shows a secondary cleavage product, with a mass of approximately 3360 Da. This corresponds to a cleavage not at the previously reported native site (Ala8-Ala9), but instead between Glu2 and Ser3. It is known that a change in membrane thickness can alter the cleavage site position[29],

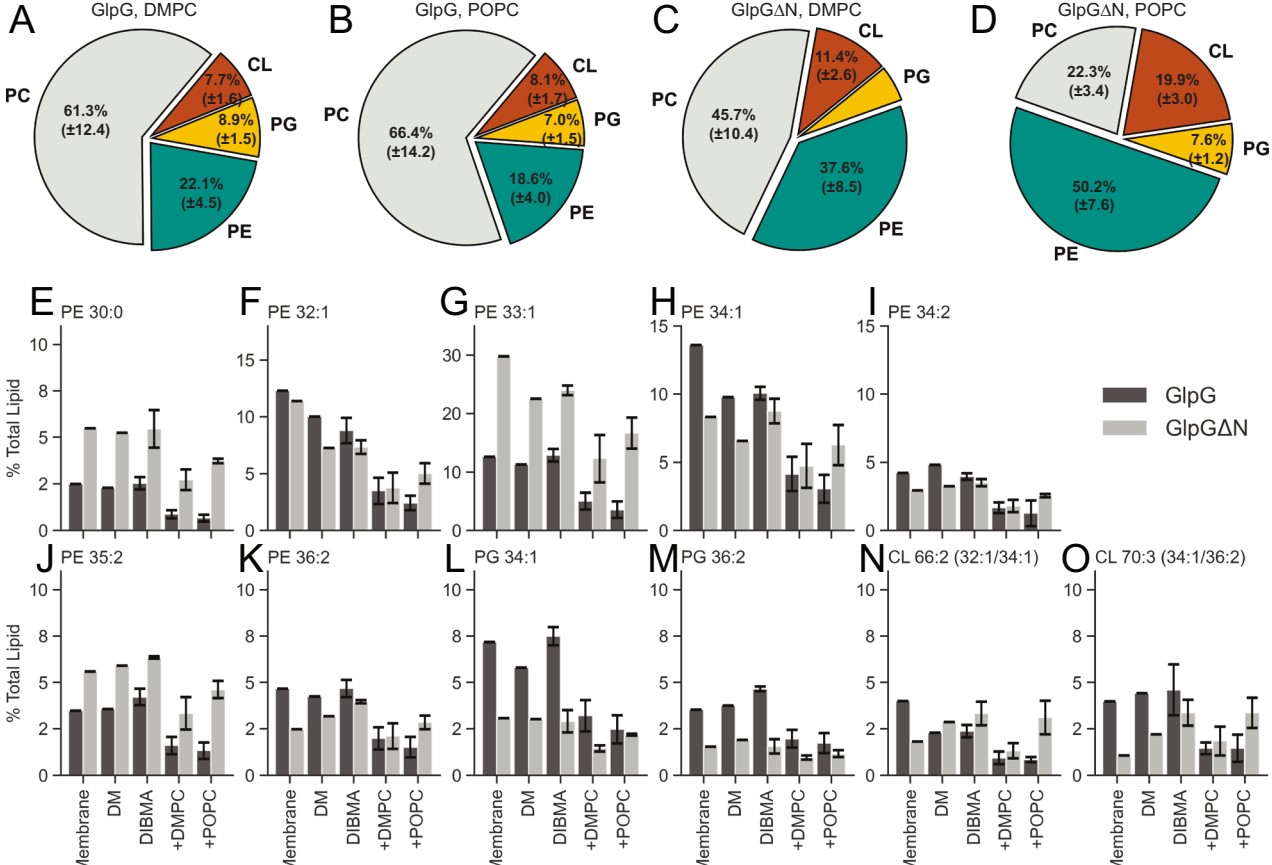

**Fig. 3 | Summary of replacement of native membrane lipids as analyzed by lipidomics.** Total lipids were analyzed by lipidomics for the post-mixing samples (DIBMALPs, **A**−**D**), and relative abundances of major lipid components within these samples (**E**−**O**). Lipid composition as a percentage of total lipid for **A** DIBMA-solubilized GlpG, post DMPC DIBMALP incubation. **B** DIBMA-solubilized GlpG, post POPC DIBMALP incubation. **C** DIBMA-solubilized GlpGΔN, post DMPC DIBMALP incubation. **D** DIBMA-solubilized GlpGΔN, post POPC DIBMALP incubation. Full values are available in Supplementary Table 1. **E**−**O** Bar charts highlighting

endogenous lipids that are present in the highest levels in both GlpG and GlpGΔN DIBMALPs, post DMPC or POPC incubation. Quantities shown are from induced membranes (C41, GlpG and C43, GlpGΔN) (Membrane), DM-solubilised (DM), DIBMA-solubilised (DIBMA), DIBMA-solubilised post DMPC incubation (+DMPC), and DIBMA-solubilised post POPC incubation (+POPC). Errors shown for **A**−**O** are standard deviation from the mean from biological replicates, where $n = 3$ for all DIBMALP containing samples, no errors are stated for DM samples ($<n = 3$). No errors are stated for membrane samples (**E**−**O**).

and that in PE-poor environments, such as PC-washed DIBMALPs, GlpG can not sufficiently thin the local membrane environment[17]. The site shifting observed for nearly all GlpGΔN samples in PC-containing samples, therefore, indicates that there is an abundance of PC lipids in the local lipid environment, as detected by the lipidomic results (Fig. 3). Finally, this shifting is only observed for GlpGΔN in PC-rich membranes, it indicates that the N-terminal deficient GlpG cannot reorient within or selectively thin the membrane in this non-native environment, but the full-length GlpG can overcome this to cleave TatA in the expected position, and at a higher rate than with only native lipids, however, the exact mechanism of this difference is unknown and requires further study.

## Discussion

Lipid-polymer nanodiscs (SMALPs/DIBMALPs) have become an increasingly popular method of membrane protein solubilization, due to the ability to retain the native lipid environment and the often-reported increased stability compared to detergent extraction methods[19]. In particular, the ability to co-extract native lipids has enabled a resurgence in lipidomics in recent years, as a viable alternative to harsher lipid extraction techniques. However, functional studies of membrane proteins within these systems have remained elusive for proteins that require membrane-bound substrates, such as the rhomboid protease family.

Previously a novel property of SMALPs/DIBMALPs has been described: the lipid composition/population of two polymer nanodiscs undergoes bulk exchange, in contrast to the singular lipid diffusion or disc fusion of peptide-based nanodiscs. This so-called 'hydrocarbon continuum' has been shown to occur in both detergent micelles and lipid-polymer nanoparticles systems[20,30,31], and facilitate a fast transfer of lipids across this continuum.

Here we show that this mixing phenomenon in lipid-polymer nanoparticles can be used to exchange the natively extracted lipid environment of membrane proteins within these DIBMALPs with non-native lipids. This allowed us to explore the different effects non-native lipids have on the activity of GlpG and GlpGΔN. Similar techniques, such as those in peptide-based nanodiscs, also exhibit this capability. These nanodiscs possess increased homogeneity over polymer systems but require detergent for their production, so they may lose some of the weaker protein-lipid associations that are of interest. Our non-invasive technique can be used to selectively introduce or wash out lipids, without the use of destabilizing detergent or organic solvents from extraction to exchange. This advantage, therefore, opens up a potential method to explore the weaker and non-specific lipid-protein interactions and examine the effects of certain lipids on membrane protein activity and stability for membrane proteins and complexes that are not stable in detergent-based systems.

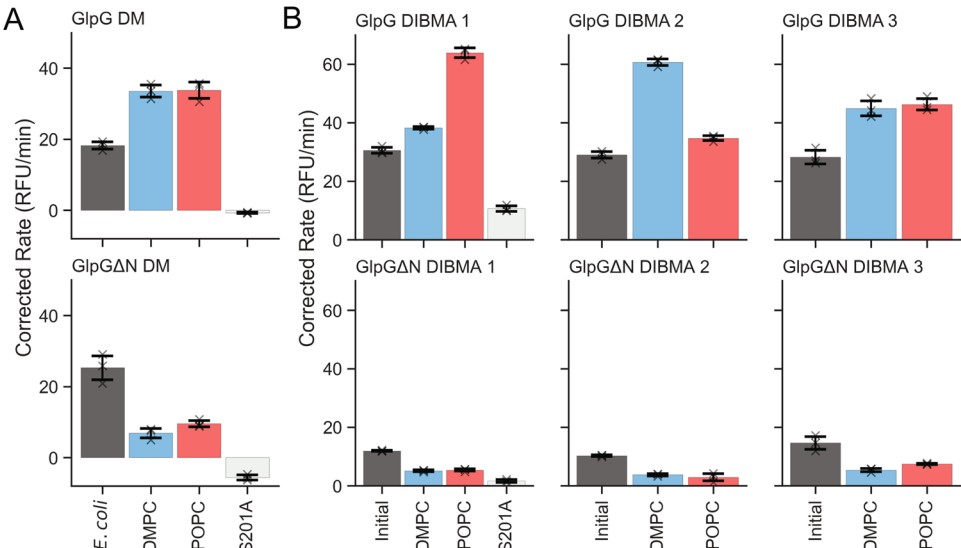

**Fig. 4 | Proteolytic activity of GlpG and GlpGΔN as measured by cleavage of TatA-FITC. A** Activity of DM-solubilized GlpG (top) and GlpGΔN (bottom) reconstituted into *E. coli* total lipid extract, DMPC, and POPC LUVs. Errors displayed derive from the standard deviation of the mean from technical repeats (*n* = 3). **B** Activity of GlpG (top) and GlpGΔN DIBMALPs (bottom) in natively extracted lipids ('Initial'), DMPC-washed ('DMPC'), and POPC-washed ('POPC') DIBMALPs.

Numbering indicates biological repeats, with errors displayed deriving from the standard deviation of the mean from technical repeats (*n* = 3), where gray crosses mark individual data points for each technical repeat. The catalytically inactive mutant S201A is represented for both DM-solubilised (**A**) and DIBMA-solubilized (**B**) as a light gray bar in the left-hand graph. Source data are provided as a source data file.

Additionally, we show through the cleavage of integral membrane protein TatA by GlpG that not only does this mixing through the 'hydrophobic continuum' occur not only for lipids, but for two protein-containing DIBMALPs, but that it occurs with no deleterious effect on the proteins involved (Fig. 5). This collisional mixing method in polymer-based nanodiscs (SMALPs/DIBMALPs) is an ideal simple methodology for minimally invasive lipid exchange and the exploration of lipid-protein association.

## Methods
### Extraction and purification of GlpG and SUMO-TatA-FLAG from *E. coli*
Both the GlpG core domain (residues 87-276, GlpGΔN) and GlpG full-length (GlpG) with a hexahistidine tag were expressed as described previously[26,32] in BL21 C43(DE3) or C41(DE3), respectively. SUMO-TatA-FLAG was expressed in a GlpG knock-out *E. coli* BL21(DE3) strain. For detergent, (n-Decyl β-maltoside (DM), Glycon) purified samples, purification followed previously described protocols[26]. For DIBMA purification, cell pellets after harvesting were weighed and resuspended to 62.5 mg/mL (*w/v*) in 62.5 mM HEPES (pH 8), 800 mM NaCl, 6.25 mM CaCl₂. DNAse and protease inhibitors (pepstatin A) were added at the recommended dosage. The cell suspension was left to resuspend for 1 h at room temperature, prior to lysis using an LM10 microfluidizer (Microfluidics, USA) with a 15,000 psi working pressure. DIBMA (Glycon), from a 10% (*w/v*) stock in 50 mM HEPES (pH 8), 100 mM NaCl, was then added to a final concentration of 2.5% (*w/v*). The resulting mixture was then incubated with agitation at 37 °C for 2 h, after which the unsolubilized membrane was removed through centrifugation at 20,000 × *g* for 30 min. The supernatant was collected and incubated overnight with buffer-incubated Ni-NTA beads at a concentration of 2 mL beads per 1 L of cultured bacteria at 4 °C. The resulting slurry was washed with 5-column volumes of wash buffer (20 mM HEPES (pH 8), 300 mM NaCl, 20 mM imidazole) and protease-containing DIBMALPs eluted in 5-column volumes elution buffer (20 mM HEPES (pH 8), 100 mM NaCl, 500 mM imidazole). Eluted fractions were then subjected to desalting using a HiPrep 26/10 desalting column to a final buffer of 50 mM HEPES (pH 8), 100 mM NaCl. Protein quantity was assessed through absorbance at 280 nm. Purity and correct folding of GlpG were checked by SDS-PAGE (Supplementary Fig. 1A, B) and solid-state NMR (Supplementary Fig. 2), respectively. Detergent purification of GlpG and GlpGΔN was performed in biological duplicates, numbered 1 and 2. DIBMA purification of GlpG and GlpGΔN were performed in biological triplicates, numbered 1, 2, and 3.

### Collisional mixing and re-purification
Both DMPC (1,2-dimyristoyl-*sn*-glycero-3-phosphocholine) and POPC (1-palmitoyl-2-oleoyl-glycero-3-phosphocholine) were purchased from Avanti Polar Lipids (Alabaster, AL) and used without further modification. For DMPC, powder was weighed whilst POPC was measured from a chloroform stock, and dried to a lipid film. Both lipid types were resuspended to a final concentration of 1 mg/mL in 50 mM HEPES (pH 8), 100 mM NaCl for 30 min at 30 °C with occasional vortexing. DIBMA was added from a 10% (*w/v*) stock in 50 mM HEPES (pH 8), 100 mM NaCl to a final amount of 1.25 times the dried lipid mass. This was then incubated for 1 h at 30 °C with occasional vortexing. To remove unsolubilized lipids, samples were centrifuged for 1 h at 180,000 × *g*, and only the supernatant was taken forward.

For collisional mixing, protein-containing DIBMALPs were incubated with a 100× molar excess of lipid with lipid-only DIBMALPs, according to protein concentration as measured by absorbance at 280 nm during desalting. The mixture was incubated for 1 h at 37 °C with gentle agitation. This was then incubated with 2 mL Ni-NTA beads overnight at 4 °C, after which protein-containing DIBMALPs were re-purified from lipid-only DIBMALPs using 5-column volumes for both the wash and elution steps. Eluted fractions were then subjected to desalting using a HiPrep 26/10 desalting column to a final buffer of 50 mM HEPES (pH 8), 100 mM NaCl. The final protein quantity was assessed through absorbance at 280 nm.

### TatA-FITC DIBMALP preparation
Briefly, a fluorescein isothiocyanate (FITC)-labeled peptide with the first 33 amino acids of TatA and a ß-alanine linker was produced (FITC-ᵦA-MESTIATAAFGSPWQLIIIALLIILIFGTKKLR) by solid-phase synthesis. The peptide was dissolved in 50 mM Tris, 150 mM NaCl,

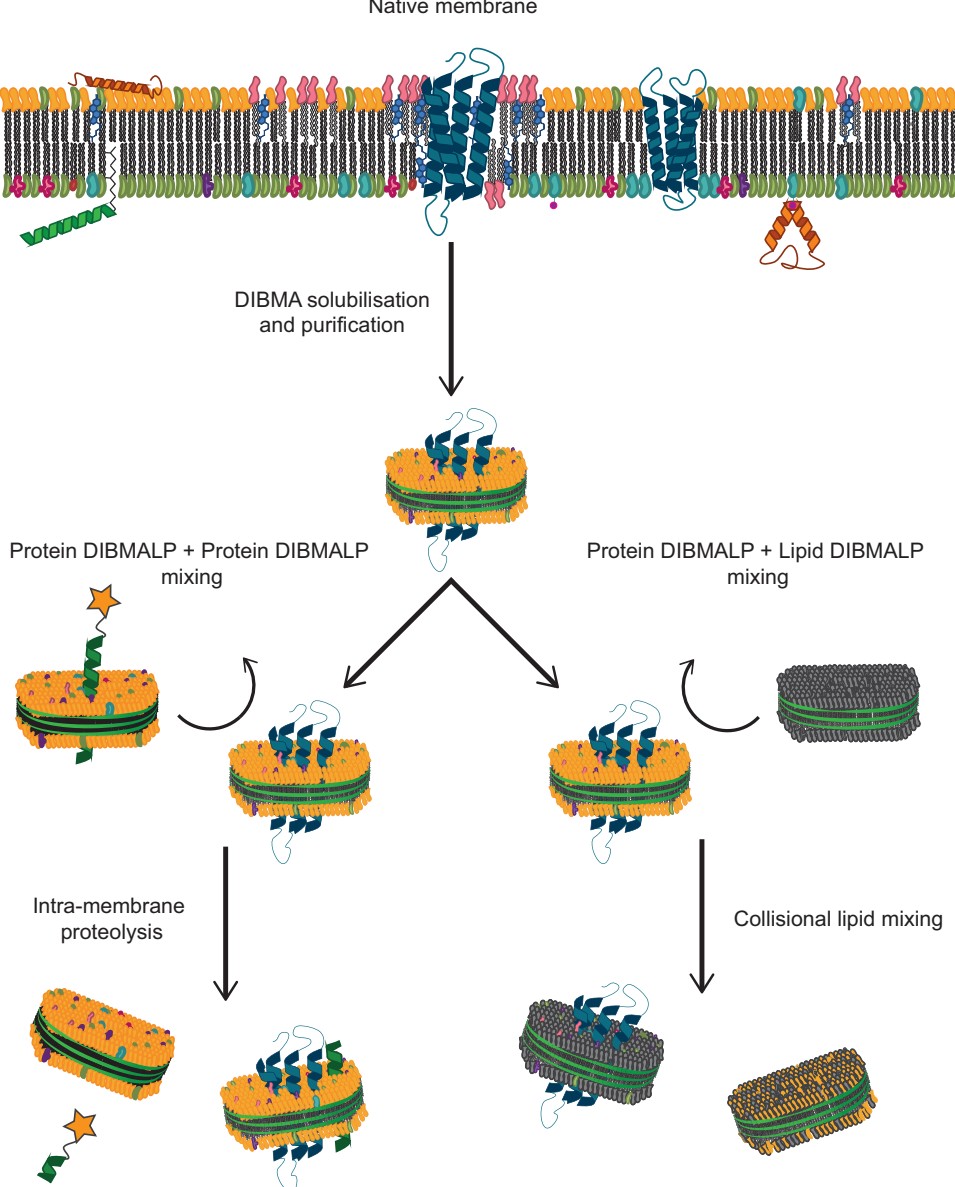

**Fig. 5 | Representative summary of methodologies presented in this work, where protein-containing DIBMALPs can be purified from native membranes, containing co-extracted native lipids.** These can then be incubated with protein-containing DIBMALPs, where intramembrane protein-protein interactions can occur (such as cleavage of SUMO-TatA-FLAG or TatA-FITC in this work), or incubated with lipid-only containing DIBMALPs, where lipid exchange can occur (such as the exchange of DMPC or POPC in this work). DIBMALPs are represented here as nanodiscs for visual ease.

and 0.2% (w/v) sodium lauryl sarcosine to a final concentration of 400 μM as described previously[27]. TatA-FITC was incorporated into liposomes of *E. coli* total lipid extract, at a molar ratio of 1:100 (peptide:lipid). The resulting proteoliposomes were solubilized through the addition of DIBMA at 1.25 × (w/w) of the original mass of *E. coli* lipid. The solution was incubated at 37 °C for 2 h, with gentle agitation. Unsolubilised material was removed through centrifugation (180,000 × g, 1 h). The peptide concentration of the resulting solution was measured through both BCA and Bradford assays.

**TatA-FITC cleavage assay (LUVs)**

Reconstitution of GlpG or GlpGΔN and TatA-FITC in LUVs was performed as described previously[27]. For POPC or DMPC LUVs, these specific lipids were substituted for *E. coli* total lipid extract but prepared in a similar manner. Fluorescence was read out every 5 min over the course of 24 h at 37 °C. Measurement was achieved using an excitation of 490 nm and detecting emission at 525 nm on a BMG Labtech

FLUOstar Omega and a sealed Corning clear flat bottom plate (#3766). The calculation of the cleavage rate employed linear regression on the linear fluorescence increase (Supplementary Fig. 4A–D), and was performed by Python script. Error determination was based on technical repeats ($n = 3$).

**TatA-FITC cleavage assay (DIBMA)**

4 μM of GlpG or GlpGΔN containing DIBMALPs were incubated with 20 μM TatA-FITC DIBMALPs to a total volume of 20 μL, and a buffer composition of 50 mM HEPES (pH 8), 100 mM NaCl. Fluorescence was read out every 5 min over the course of 24 h at 37 °C. Measurement was achieved using an excitation of 490 nm and detecting emission at 525 nm on a BMG Labtech FLUOstar Omega and a sealed Corning clear flat bottom plate (#3766). The calculation of the cleavage rate employed linear regression on the linear fluorescence increase (Supplementary Fig. 4E–J), and was performed by Python script. Error determination was based on technical repeats ($n = 3$).

## MALDI-TOF of TatA-FITC cleavage products

Supplementary Figs. 5 and 6 were generated using similar methods as stated for the TatA-FITC cleavage assays. For DM-only samples (Fig S5A–F), 4 μM GlpG or GlpGΔN in 50 mM Tris pH 7.4, 300 mM NaCl, 10% glycerol, 0.2% (w/v) DM was added to 20 μM TatA-FITC in 20 mM HEPES pH 7.3, 150 mM NaCl and 0.2% (w/v) sodium lauryl sarcosine. All samples were incubated at room temperature with gentle agitation for between 16 and 24 h.

For MALDI-TOF preparation, DM only samples were applied to the MALDI sample plate in a 1:1 (v/v) mixture of sample and matrix (composed of α-Cyano-4-hydroxycinnamic acid saturated solution of 70:30 ACN:H$_2$O). LUV and DIBMA samples were pre-incubated with a 1:1:1 (v/v/v) final volume of sample:2% TFA in H$_2$O:10% DM in H$_2$O for at least 1 h to enable sufficient ionization. These were then deposited onto MALDI plate in either a 1:1 (v/v) sample:matrix mixture, or 1:2 sample:matrix mixture to increase signal:noise. Spectra were then exported to be analyzed by Python script.

## Quantitative mass spectrometry of lipids

Mass spectrometric analysis was performed essentially as described previously[33,34]. Lipid extraction from nanodiscs was performed according to Bligh and Dyer, with modifications. Briefly, an estimated 5 nmol total lipid in 240 μL water and internal standards (1.25 μl EquiSPLASH and 6.25 μl cardiolipin (CL) mix I, Avanti Polar Lipids, corresponding to a standard lipid mix molar amount of: 18:1(d7) Chol Ester (76 pmol), 15:0-18:1-d7-PC (66.4 pmol), 15:0-18:1-d7-PE (70.3 pmol), 15:0-18:1-d7-PI (59 pmol), 15:0-18:1-d7-PS (64.4 pmol), 15:0-18:1-d7-PG (65.44 pmol), C15 Ceramide-d7 (94.2 pmol), d18:1-18:1(d9) SM (67.8 pmol), 15:0-18:1(d7) DAG (85 pmol), 15:0-18:1(d7)−15:0 TAG (61.6 pmol), CL 57:4 (40.41 pmol), CL 61:1 (36.9 pmol), CL 80:4 (33.86 pmol), CL 86:4 (30.82 pmol)) was mixed with 0.9 mL of chloroform/methanol (1:2 (v/v)) for 10 min. After addition of 0.3 mL chloroform and 0.3 mL H$_2$O, the sample was mixed again for 10 min, and phase separation was accelerated by centrifugation (1000 × g, 3 min). The lower chloroform phase was carefully transferred to a clean glass vial. The upper water phase was mixed with 10 μL 1 M HCl and 300 μL chloroform for 10 min. After phase separation, the lower chloroform phase was carefully transferred to the glass vial with the chloroform phase from the first extraction. The solvent was evaporated by a gentle stream of argon at 37 °C. Lipids were dissolved in 10 mM ammonium acetate in methanol, transferred to Twin.tec PCR plate sealed with Thermowell sealing tape and analyzed on a QTRAP 6500 triple quadrupole mass spectrometer (SCIEX) equipped with nano-infusion splay device (Tri-Versa NanoMate with ESI-Chip type A, Advion). The quadrupoles Q1 and Q3 were operated at unit resolution. PC analysis was carried out in positive ion mode by scanning for precursors of m/z 184.1 at a collision energy (CE) of 37 eV. PE and PG measurements were performed in positive ion mode by scanning for neutral losses of 141 Da and 189 Da at CEs of 25 eV and 30 eV, respectively. CL species were identified in positive ion mode by scanning for precursors of the masses (m/z 465.4, 467.4, 491.4, 493.4, 495.4, 505.5, 519.5, 521.5, 523.5, 535.5, 547.5, 549.5, 551.5, 573.5, 575.5, 577.5, 579.5, 601.5, 603.5, 605.5, 607.5, 631.5, 715.5 and 771.5 Da) corresponding DAG-H$_2$O fragments as singly charged ions at CEs of 40–50 eV. Mass spectra were processed by the LipidView Software Version 1.2 (SCIEX) for identification and quantification of lipids. Lipid amounts (pmol) were corrected for response differences between internal standards and endogenous lipids. Correction of isotopic overlap in CL species was performed as described previously[26]. MS/MS scans to determine acyl-chain compositions in PE and PG were performed in negative ion mode by scanning for precursors of the masses of acyl chains (m/z 253.2, 255.2, 267.2, 269.2, 281.2, 283.2, 295.2) at a CE of −45 eV. The raw data and mass spectra for samples discussed in this work can be found at the https://doi.org/10.5281/zenodo.7778567.

## Solid-state NMR

($^{13}$C,$^{15}$N)-labeled GlpG was expressed in the same way as unlabeled GlpG but with uniformly $^{13}$C labeled glucose as the sole carbon source and $^{15}$N labeled ammonium chloride as the sole nitrogen source. After extraction and purification into DIBMALPs (as described above) rotor packing was achieved through ultracentrifugation. The sample (of approximately 1–2 mL in volume) was spun for 36 h at 100,000 × g into a 1.9 mm rotor, in a similar manner as previously described by Bersch et al.[35]. The NMR sample of GlpG in E.coli total lipid extract liposomes (lipid:protein ratio of 32:1) was prepared as previously described[26]. Solid-state NMR spectra were recorded on 600 MHz (liposome sample) and 700 MHz (DIBMA sample) spectrometers (Bruker) equipped with 1.9 mm probes, operating at magic-angle spinning (MAS) rates of 35 kHz (DIBMA sample) and 40 kHz (liposome sample). The sample temperatures were 20 °C (DIBMA sample) and 23 °C (liposome sample), determined from the chemical shift of the water peak in a $^1$H 1D spectrum referenced to Sodium 3-(trimethylsilyl)propane-1-sulfonate (DSS). The 2D hNCα spectrum of the DIBMA sample was recorded with 4096 scans, 15 ms acquisition time in the direct dimension ($^{13}$C), and 11.9 ms acquisition time in the indirect dimension ($^{15}$N). $^1$H-$^{15}$N cross-polarization (CP) was achieved using a ramp (100−80%) with a maximum amplitude of 72 kHz on the $^1$H channel, a rectangular pulse with a nutation frequency of 30 kHz on the $^{15}$N channel, and a mixing time of 1 ms. $^{15}$N-$^{13}$C cross-polarization (CP) was achieved using a ramp (80−100%) with a maximum amplitude of 26 kHz on the $^{13}$C channel, a rectangular pulse with a nutation frequency of 14 kHz on the $^{15}$N channel, and a mixing time of 5.5 ms. The 2D hNCα spectrum of the liposome sample was recorded with 4096 scans, 15 ms acquisition time in the direct dimension ($^{13}$C), and 13.1 ms acquisition time in the indirect dimension ($^{15}$N). $^1$H-$^{15}$N cross-polarization (CP) was achieved using a ramp (80−100%) with a maximum amplitude of 70 kHz on the $^1$H channel, a rectangular pulse with a nutation frequency of 12 kHz on the $^{15}$N channel, and a mixing time of 0.65 ms. $^{15}$N-$^{13}$C cross-polarization (CP) was achieved using a ramp (80−100%) with a maximum amplitude of 23 kHz on the $^{13}$C channel, a rectangular pulse with a nutation frequency of 13 kHz on the $^{15}$N channel, and a mixing time of 7 ms. Low power 1H decoupling (10 kHz during acquisition), no decoupling during the N-C cross-polarization transfer, and short recycle delays (1 s for the DIBMA sample and 1.2 s for the liposome sample) were possible due to the fast MAS rates. The spectra were processed with TopSpin 4 (Bruker) and visualized using CcpNmr Analysis V.3[36].

## Reporting summary

Further information on research design is available in the Nature Portfolio Reporting Summary linked to this article.

## Data availability

The Lipidomics data generated in this study have been deposited in Zenodo, with the accession code 7778567. The raw data for fluorescence and MALDI data is available within the Source Data file. NMR spectral data and any other data is available upon request. Source data are provided with this paper.

## Code availability

Code used to analyze the data presented in this work is available upon request.

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

## Acknowledgements

We thank Natalja Erdmann and Susanne Bischoff for their expert tech-nical assistance, we would further like to thank Ines Kretzschmar for peptide synthesis expertize. This work was supported by the Leibniz-Forschungsinstitut für Molekulare Pharmakologie (FMP) and the Deutsche Forschungsgemeinschaft (DFG, German Research Founda-tion) under Germany´s Excellence Strategy—EXC 2008/1 (UniSysCat)—390540038 (to A.L.). C.Ö. acknowledges funding from the Human Frontier Science Program LT000303/2019-L.

## Author contributions

H.S. produced samples for lipidomic analysis, fluorescence measure-ments, MALDI-TOF data, and gel imagery, as well as analyzed generated data. T.T. and T.L. performed lipidomic data acquisition and advised on sample preparation. C.Ö., S.K., and A.L. performed solid-state NMR mea-surements and analysis. S.L. oversaw sample production and advised on experimental procedures. C.B. performed fluorescence measurements. A.L., T.L., and H.S. designed this study. H.S., A.L., T.T., and C.Ö. wrote the manuscript, with other authors providing advice and proof-reading.

## Funding

## Competing interests

The authors declare no competing interests.
