## [Peer Review File · Nature Communications]

Lipid-polymer nanoparticles to probe the native-like environment of intra-membrane rhomboid protease GlpG and its activityReviewers' comments:

Reviewer #1 (Remarks to the Author):

The manuscript submitted by Sawczyk et al. characterizes the rhomboid protease GlpG in lipid nanodiscs formed by the copolymer DIBMA and the exchange of lipid and protein between these complexes. While mixing of lipids between DIBMA nanodiscs has previously been established, this is the first study to show that lipid exchange can also be achieved when the nanodiscs are loaded with protein, and that the proteins themselves can be transferred between nanodiscs. The latter result is of particular interest because it provides a method for intramembrane protease function to be evaluated without exposure to detergents normally used to solubilize substrate and protease. This allows assays of intramembrane protease activity to be done in more native-like environments than was previously possible. Quantitative analysis of lipid composition for these nanodiscs and comparison with native membranes was also done to investigate potential interactions with specific lipids in GlpG nanodiscs, and the role of the N-terminal cytoplasmic domain in these interactions. Protease activity was found to be affected by nanodisc composition, in line with previous studies on GlpG and its N-terminal truncation GlpG Δ N.

Although this is an interesting study, there are significant aspects of this work that undermine the validity of some of the conclusions that have been made. These are outlined in the following points:

1) Sample purity and protease assay interpretation

The purity of GlpG samples does not seem to be very high, which can complicate interpretation of protease assay results. Gel analysis of purified GlpG (Fig. S1) shows many contaminants, and purity levels that I would estimate to be 50% or less for the GlpG samples prepared in native membranes. While it is not surprising that GlpG in lipid nanodiscs from native membranes would co-purify with other proteins, it is misleading to state that the purity was confirmed (page 5, line 213) with no comment on the relatively low level of purity in the native membrane nanodiscs. Also, it is interesting that the purity of samples in Figure 4A appear to be significantly higher than those shown in Figure S1 A and B. What is the difference between these samples? On the positive side, the purity is significantly improved after lipid exchange, which likely reflects the impact of diluting non-tagged contaminants into separate nanodiscs followed by another nickel affinity chromatography step. This seems to be a feature worth mentioning in the manuscript.

Related to sample purity is a solid-state NMR spectrum (Fig. S2) that was offered as evidence for sample homogeneity. The spectrum shows the presence of folded protein, however the gels show a number of proteins are present in the sample, some at significant concentrations. With no

comparison to previously published spectra it is not possible to know if GlpG alone is the source of peaks observed in this spectrum, particularly if contaminants have more favorable relaxation properties than GlpG.

The presence of contaminating proteins in the purified nanodiscs can affect assays of protease function should any of these contaminants also have protease activity. In particular, release of the fluorescent tag from TatA-FITC could result from the action of contaminating proteases cleaving aqueous-exposed residues. It is encouraging that rhomboid-specific activity is strongly corroborated by MALDI-MS showing that at least one of the cleavage products corresponds to that expected for TatA-FITC cleaved by GlpG. However, the release of fluorescence may happen at a faster rate than GlpG-mediated cleavage, a possibility that can't be ruled out without a negative control (ideally the same experiment run on an active site serine knockout mutant). Non-GlpG protease activity could explain the difference in rates observed between the fluorescence-based assay (minute time scale) and the gel-based assay (days). The presence of other proteases could also account for the disappearance of a contaminant band in the gel assay seen around 30 kDa in the FLAG-TatA-SUMO sample following incubation with GlpG (Fig. 4A). In addition, there is a difference in cleavage products between the two GlpG samples in this gel analysis, with a significant band around 13 kDa present in the GlpG cleavage not seen in the GlpG Δ N reaction. Differences in copurification of contaminating proteases could also explain this. Taken together, all of these discrepancies raise significant questions about the activity being measured in these assays – how much is GlpG-specific? The use of a negative GlpG control in all of these experiments, with the active site serine knocked out, is required to establish that the measured activity was due to GlpG, and not co-purified contaminants.

2) Difficulties with comparisons of lipid compositions to monitor protein-lipid interactions

Lipid compositions were analyzed to investigate differences between lipid interactions with GlpG and GlpG Δ N. These two proteins were expressed in two different strains of *E. coli*, each having a different lipid composition in the membrane extracts. Although the authors suggest that the differences in lipid composition observed between nanodiscs with GlpG versus GlpG Δ N are due to differences in protein interactions with lipids, it is difficult to deconvolute the influence of GlpG on these compositions from the differences in the native membrane compositions that each sample was extracted from. It makes more sense to this reviewer to compare the composition of lipids between samples all prepared from the same strain. However, by this analysis a different set of conclusions could be drawn in place of the conclusions provided in the manuscript. For example, GlpG in DDM or DIBMA showed enrichment of cardiolipin relative to levels observed in lipid extracts. This would appear to occur at the expense of PG and PE, a trend that was observed for both full length and truncated GlpG. One might conclude that there are specific interactions between GlpG and CL, although as the authors note, this also reflects differences in tendency for these lipids to be incorporated into DDM micelles or DIBMA nanoparticles. Nonetheless, this conclusion is different from the conclusion presented in the manuscript based on a comparison of

lipid compositions of nanodiscs with GlpG from C41 versus GlpG Δ N from Tuner E. coli suggesting a specific interaction with PE. These different possible conclusions highlight the complications that arise when multiple variables are changed simultaneously. More reliable conclusions could be made if the analysis was done using both GlpG and GlpG Δ N expressed in the same E. coli strain, or with DIBMA nanodiscs prepared from the two E. coli strains without expressed GlpG. (I am assuming that the figure legend for figures 2A and 2D is not correct. Based on what was written in the discussion it seems like these are lipid compositions of membrane extracts that have not been solubilized by DIBMA. Whatever these samples are, the legend needs to accurately describe them, and the preparation of these important reference samples should be described in the Materials and Methods.)

3) Experimental uncertainty and significant differences

There is no indication of experimental uncertainty for any of the percent lipid composition measurements, which makes it impossible to evaluate the significance of differences between samples. This is an issue throughout the analysis, for example, on page 6, line 294, a conclusion about the native lipid PE 33:1 is made that “This retention indicates a stronger, or preferential, binding interaction than normal for a protein with bulk membrane”. Yet it is not clear what levels would be expected for these species if there was no binding interaction. If the levels of retained PE 33:1 are deemed to be above background levels, then by the same metric it seems like the amount of cardiolipin contained in these lipid-diluted nanoparticles would also be significant, yet this was not identified. The absence of experimental uncertainty prevents objective assignment of significant versus insignificant differences, making it difficult to understand the rationale for the conclusions being presented.

4) Cardiolipin levels

One point of curiosity is the levels of cardiolipin being detected in this study. These are very high compared to the values that are previously seen in E. coli membranes expressing GlpG i.e. <5% (Reading et al, Angew Chemie Int Ed 2017). Why is there such a large amount of cardiolipin in these samples?

In addition to the major issues outlined above, there are also some changes that should be made to clarify minor points:

Page 1, line 50 – Not all substrates have proline within 4 – 5 residues of cleavage site (e.g. LacY TM2). Also, not before cleavage site, but after cleavage site.

Page 2, line 64 - Wording of this sentence is not clear. Should clarify that exchange of lipid between peptide-based nanodiscs has been demonstrated, albeit by a different mechanism where lipid monomers are released from the complex and incorporated into a different nanodisc.

Page 2, line 68 – Would be better (more clear) to replace ‘previously reported lipid-only’ mixing with ‘collisional mixing of pure lipid nanodiscs’.

Page 2, Figure 1 – The resolution of the figure in the reviewer’s copy is low, making it difficult to distinguish 2 lipid colours in lipid nanodiscs.

Page 2, line 97 – What is a self-generated knock out strain?

Page 5, line 222 – Clarify what is meant by “membrane-only fractions for both Tuner and C41 cells”. Are these total membrane extracts from cells expressing GlpG? Or are they DIBMA-extracts from cell strains that are not expressing GlpG?

Page 5, line 228 – When talking about ‘enrichment’ the identity of the reference needs to be clarified. Enriched relative to membrane extracts of cells expressing GlpG? Enriched relative to DIBMA-extracted lipids from cells that do not express GlpG?

Reviewer #2 (Remarks to the Author):

The manuscript by Sawczyk and colleagues addresses the use of DIBMA nanodiscs to test the effect of the immediate lipid environment on rhomboid membrane protease GlpG.

Intramembrane proteases are unusual proteases because enzymatic water-dependent catalysis occurs in the hydrophobic space of a lipid bilayer. The enzymes are involved in a variety of biological processes, for example gamma-secretase and SPP and SPPL2a-c proteins are among them, which are highly relevant from a biomedical point of view. Thus, understanding the molecular mechanisms that contribute to regulation of intramembrane protease activity are highly relevant from a basic and applied research perspective.

The role of the direct lipid environment in controlling intramembrane protease activity is poorly understood. One reason for this is that the intramembrane proteases are difficult proteins, both in terms of expression and purification, and detailed analysis of the cleavage products. Furthermore, there is still a lack of assays that allow a systematic and controlled analysis of the influence of lipids on the structure and function of these enzymes.

Lipid polymer nanodiscs have been shown to be very suitable for studying the lipid remodeling of membrane proteins. In the present manuscript, the possibility of collisional lipid mixing is used to systematically investigate the influence of lipids - and specifically the controlled exchange of the lipid environment - on rhomboid protease activity. As a proof of principle, the authors investigate the role of the lipid environment on the protease activity of the full-length protein in comparison to truncation variants of GlpG and report that different dependencies on the lipid environment exist for the enzymatic properties of the two proteins.

The presented system has the potential to provide new insights into understanding the role of lipids in regulating activities of membrane proteins. However, the manuscript could have benefited from a clear decision as to whether it should be a methodological paper or a results-oriented paper. In my opinion, as a methodological paper it lacks a number of controls and as a results-oriented paper it lacks data that go substantially beyond the findings already published.

Comments

In general, the experiments, results (including legends) and their interpretation are sometimes explained incomprehensibly or incompletely.

How often were the experiments shown in Fig 2 and 3 performed and were different protein preparations used for this purpose? There is no indication that the experiments were performed more than once.

Regarding the quality of protein preparations: I might have missed it but what is the difference between the protein preparations shown in Figure 4A compared to those in Fig S1 and S2. Figure S1, which is intended to demonstrate the purity of the proteins used, shows a contaminant in DLpG deltaN running at the height of the full length protein. Western blots using anti-his antibody with GlpG and GlpG deltaN on one gel should be added. The authors state in lines 213ff that for GlpG sample homogeneity was confirmed by NMR. Here the contamination (if there?) in the GlpG deltaN preparation should be mentioned. Intact mass and tryptic digest data should be added. Protein quality and purity is an important point when it comes to interpretation of results obtained for co-purified lipids and enzyme activities.

In lines 317ff, the authors state that substrate cleavage in their system takes 7 days instead of 12-16 hours reported for other systems. The authors should discuss what imitations this entails.

Lines 265ff: What about the results reported by Almeida-Hernandez (2019) on the role of PG in membrane thinning. This should be discussed here.

Fig S4B versus D: The different kinetics of GlpG versus GlpGdeltaN should be discussed.

The methods section should be expanded, the authors often only refer to previously published literature or miss to give references (see lines 98ff). Some aspects are not described, e.g. TAMRA-FP labeling and LUV reconstitutions.

Line 81, The term 'washed' might be misleading.

Figure 1: Why not add the substrate if this is a novel aspect of the method?

Lines 227: I suggest to rephrase this sentence as the effect of 'disfavoring PE lipids by DIBMA is more pronounced for GlpGdeltaN than for GlpG.

Fig S3: The top spectra are cut off in the upper part.

Reviewer #3 (Remarks to the Author):

The paper by Sawczyc et al reported the use of DIBMALP to probe the "native environment" of GlpG and its activity by means of "collisional mixing". This paper takes for granted many unsubstantiated claims in the SMALP literatures and tries to explain its observations by aligning with some of the previous claims. In my opinion, this is a missed opportunity as the authors are better off building hypothesis based on their own observations and testing it, which will truly enrich our understanding of DIBMALP (or SMALP in general). I would like to raise a few questions for the authors to consider:

(1) The authors refer "lipid-polymer nanoparticles" as SMA or DIBMA. This is incorrect because SMA or DIBMA refers to the amphiphilic copolymer frequently used to produce "lipid-polymer nanoparticles", not the "lipid-polymer nanoparticles" themselves. The "lipid-polymer nanoparticles" should be called "SMALP" or "DIBMALP" if we follow the nomenclature proposed by Overduin and colleagues.

(2) Related to (1), the authors refer the "lipid-polymer nanoparticles" produced by solubilizing membranes with DIBMA copolymers as "DIBMA nanodiscs". This is a misleading statement frequently (unfortunately!) used by many researchers in the SMALP (or DIBMALP) community. SMALPs (or DIBMALPs) are "lipid-polymer nanoparticles" that cannot be simply regarded as "nanodiscs". In fact, it has never been explicitly proved that SMALPs (or DIBMALPs) products are all "nanodiscs", but this hypothesis has gained popularity anyway partly because many researchers in

the field credulously repeat the hearsay without critically analyzing their own data. I'd like to point to the authors recent studies using SAXS (doi.org/10.1021/acs.langmuir.1c00304) and a set of comprehensive analytical tools (doi.org/10.1021/acs.biomac.3c00034) that clearly revealed the structural heterogeneity in SMALPs (or DIBMALPs) that debunked the wrongful simplification of "SMA nanodiscs" or "DIBMA nanodiscs". I would recommend the authors to stop using "DIBMA nanodiscs" as this wrongful concept misleads the audience.

(3) The authors repeatedly made statement about "DIBMA nanodiscs with native lipid environment". To the authors defense, the claim that SMALPs (or DIBMALPs) are "native nanodiscs" have been frequently reported in SMALPs (or DIBMALPs) literatures. This is unfortunately another bogus hypothesis in the SMALPs (or DIBMALPs) field that has never been truly substantiated. We need be serious when we make claims about "native lipid environment". To me, "native lipid environment" means both the lipid composition and organization in SMALPs (or DIBMALPs) are not altered in comparison with their parental membranes, neither of which could not be more wrong from reality. Unless proved true, the wishful thinking that SMA (or DIBMA) acts as "cookie cutter" to dissect native membrane without altering it has greatly misled the community and needs to be resolutely refuted in future publications. For example, lipidomic analysis (ref 27) clearly showed that that lipid compositions in SMALPs (or DIBMALPs) differ from the native membrane and from each other, and recent study (doi.org/10.1021/acs.biomac.3c00034) clearly showed that intercalation of SMA alters lipid-lipid and lipid-protein interactions. The authors need not look further but their own lipidomic analysis in Figure 2 – does it truly support the "native lipid environment" of DIBMALP? By the way, many studies made claims about "native lipid environment" of SMALP (or DIBMALP) based on the detection of a few endogenous lipids in SMALPs (or DIBMALPs). If we follow this logic, we might as well call detergent-solubilized membranes "native micelles" as they too contain endogenous lipids. In fact, the authors own lipidomic analysis (Figure 2) clearly indicates the co-extracted lipids by DDM resemble more closely to the composition of lipids in native membranes than those extracted by DIBMA. I would recommend the authors to stop repeating the wrongful claim of "DIBMA nanodiscs with native lipid environment" as it grossly misleads the audience.

(4) I would recommend the authors to be more cautious when make claim that detergent solubilization would delipidate the "native lipids" associated with GlpG while DIBMA would not (which is by the way another popular overly-simplified claim frequently seen in literatures). Detergents comes in many different flavors with different detergency, so does the DIBMA chains (note DIBMA is a mixture of polymer chains with different hydrophobic/hydrophilic repeating unit ratio, sequence, and molecular weight). In fact, the authors own data (Figure 2 & 4) clearly indicate DDM-solubilized GlpG doesn't delipidate GlpG.

(5) Another common misconception is about the somewhat magic power of native lipids on activating membrane proteins. While the important role of native lipids on the activation of some membrane proteins have been unambiguously shown, membrane proteins are such a diverse

family and we should also recognize the fact that many membrane proteins are perfectly active in non-native lipid environment, like the authors observation of increased GlpG activity in POPC or DMPC membranes. The authors tried to explain the GlpG activity by proposing that “DDM-solubilization not leading to total delipidation, causing some native lipids from E. coli being transferred to the POPC or DMPC LUVs during protein reconstitution”. Not only is this claim unsubstantiated (no experiment is presented to test this claim), it is also a missed opportunity to really understand what it takes to activate GlpG.

(6) The most important contribution of this work is the discovery that GlpG solubilized in DIBMALPs shows protease activity to its substrate, but this could only happen when GlpG and its substrate are co-localized. There could be many different hypothesis to explain why this happens at all, and the authors proposed “collision cargo exchange” as the mechanism following Keller’s earlier proposal of the “collision lipid exchange” model, with the exception that no evidence was provided to support the “collision cargo exchange” mechanism. The authors also did not make it clear what is the “cargo” in the exchange? The lipids, the substrate, or the GlpG? By the way, lipid exchange among DIBMALPs or SMALPs or detergent micelles in general occurs naturally through the dynamic equilibrium of lipids in the self-assembled nanostructures with that freely dissolved in solution, and this is nothing new. Keller’s earlier contribution (ref 18) is that they established kinetic models to describe the two routes (i.e., freely diffusion vs collision exchange) and measured the exchange rates to show that different exchange rates exist, and the exchange rates are dependent on ionic strength and SMALP sizes (doi.org/10.1016/j.eurpolymj.2018.09.043). This observation indirectly supports the possibility of a “collision exchange” model, or as Keller puts it, the collisional exchange dominates at millimolar lipid concentrations, whereas diffusional transfer dominates at lower concentrations. This is an interesting hypothesis rather than a slam dunk conclusion that everyone would agree. In this context, the authors need make clear what “collision cargo exchange” means in their hypothesis, and design experiment to test their hypothesis. Without that data, I can’t recommend proposing of this new concept as it is no more than another scientific fantasy.

Reviewer #4 (Remarks to the Author):

Studies have shown the importances of avoiding detergents and using near-native membrane mimetics such as nanodiscs in the functional reconstitution of membrane proteins and further structural studies.

In particular, synthetic polymers are increasingly used to develop nanodiscs with various size and stability under different conditions including pH, T, and the presence of divalent metal ions. One of the unique properties of these nanodiscs is their ability to collide and exchange lipid contents. The rate of collision, stability, and stability-size-homogeneity have been shown to vary with the type of belt (and charge) and the lipid composition. In this study, the authors well utilized this unique property to demonstrate the lipid environment within the DIBMA-based nanodiscs can be exchanged with non-native lipids. In addition, this property is effectively used to explore the

different effects non-native lipids have on the activity of GlpG and GlpG Δ N. These are exciting results and will be useful to researchers in the field. The manuscript is well written. Therefore, I strongly recommend the publication of the manuscript after revision.

1) While lipid-exchange is an useful property of the polymer or peptide based nanodiscs, it can be a limitation if size homogeneity and stability are desired. This should be mentioned.

2) The authors should briefly mention and include the studies that demonstrated (using high-speed AFM and ³¹P NMR in real-time) the lipid-exchange process (Chem. Mater. 2018, 30, 10, 3204) and its use to determine the type of lipids preferred by membrane-bound cytochrome proteins (Angew Chem Int Ed Engl. 2018, 57(13):3391; Chem.Comm. 2018, 54(49):6336).

3) A recent study based on non-ionic (inulin based) polymer demonstrated that the nanodiscs collision and fusion can result in size heterogeneity. But, it can be overcome by tuning the charge of the lipids encased.

Firstly, we would like to thank the reviewers for their in-depth and constructive feedback. We have taken this to heart and hope the changes made with resubmission answer the questions put forward. The paper we have resubmitted has changed a fair amount due to this feedback so the general changes have been broken down below, before individual responses are given to specific issues brought up by the reviewers:

Comparisons between full-length (wild type) GlpG and GlpG Δ N

The strains used initially were BL21 C41 (DE3) for GlpG full-length (F.L), and Tuner for GlpG Δ N. These are the historically used strains for protein production for NMR in our group. Due to the relatively large amount of sample required for the initial purification and subsequent washes, including activity assays and lipidomics, this initial strain combination was used, as previous strain-checking and expression optimisation efforts for NMR sample preparation were already performed. After review comments, these strain checks were revisited and expanded upon with the strains BL21 (DE3), BL21 Star (DE3), BL21 pLys (DE3), BL21 C41 (DE3), BL21 C43 (DE3), and Tuner being tested (Figure 1, this text). Sadly, only BL21 (DE3) and BL21 pLys (DE3) showed comparable expression for both F.L and Δ N in small-scale expression tests, however scaled up expressions (500mL-1L) showed that only one or the other protein expressed at the required level for further experimentation (Figure 1 B and C, this text). However, in light of trying to keep the strains as similar as possible, it was possible to express the GlpG Δ N protein in the more similarly related BL21 C43 (DE3) strain (Figure 1A, this text). This is the strain that was used for GlpG Δ N, including the S201A mutant. The full length S201A was expressed to satisfactory levels in BL21 C41 (DE3), the same as wild type GlpG full length.

Figure 1 – Strain checks and expression tests performed in attempt to gain same-strain expression of GlpG F.L and Δ N. (A) Small-scale purification from 50mL test expression of commonly used strains for NMR production, showing strong C41 expression for GlpG F.L and strong expression for GlpG Δ N in C43

and Tuner strains. Weak expression was observed once for C41 for GlpG Δ N, but this was not observed repeatedly, and would have been to insufficient level if scaled-up. (B) IMAC chromatogram of positive small-scale clones of GlpG F.L (top) and Δ N (bottom) in BL21 pLys (DE3) in 500mL cultures, showing successful expression only in GlpG F.L. (C) IMAC chromatogram of positive small-scale expression test clones of GlpG F.L (left) and Δ N (right) in BL21 (DE3) after 10 hours of expression time.

Sample purity and negative controls

Sample purity for some initial purifications were indeed non-optimal, but now slight modification of the purification procedure has enabled a vast increase in sample purity, seen in Fig. S1 A and B. Similarly, a trypsin digest from contaminant bands has been performed, and shown that none of the non-GlpG bands contain a known enzyme that could affect the recorded activity or potentially lipidomic results (Fig. S1). On a similar note, the inactive mutant of S201A was included as a negative control for activity, for both F.L and Δ N samples. As can be seen in Fig. 4, it shows the expected negligible activity, again showing that the contaminants observed in the gels (Figure 2, this text) do not affect the observed activity results.

Figure 2 – Highlighted SDS-NuPAGE gel from Fig. S1 showing regions extracted for trypsin digest and MS analysis to identify protein bands. (A) Full length GlpG initial purification, showing samples 1-6. (B) DM-extracted GlpG showing samples 1-3. The full description of contaminants can be found in the Fig. S1 legend.

Uncertainty and sample variation

Since submission we have now expanded the experimental work to be not only in recorded triplicate, but in biological triplicate (for DIBMA-solubilised F.L and Δ N), as well as biological duplicate for DM-solubilised F.L and Δ N). We believe this confirms the method of lipid mixing when it comes to activity measurement, as well as the analysis of the lipidomic data. The lipidomic workflow and analysis itself has been further optimized for *E. coli* lipids, with enhanced internal standards being used which we believe to be now closer to the actual values of CL and PE/PG.

We once again thank the reviewers for their in-depth questions, and will answer the more reviewer-specific queries below, where relevant in the changed paper draft.

Reviewers' comments:

Reviewer #1 (Remarks to the Author):

The manuscript submitted by Sawczyc et al. characterizes the rhomboid protease GlpG in lipid nanodiscs formed by the copolymer DIBMA and the exchange of lipid and protein between these complexes. While mixing of lipids between DIBMA nanodiscs has previously been established, this is the first study to show that lipid exchange can also be achieved when the nanodiscs are loaded with protein, and that the proteins themselves can be transferred between nanodiscs. The latter result is of particular interest because it provides a method for intramembrane protease function to be evaluated without exposure to detergents normally used to solubilize substrate and protease. This allows assays of intramembrane protease activity to be done in more native-like environments than was previously possible. Quantitative analysis of lipid composition for these nanodiscs and comparison with native membranes was also done to investigate potential interactions with specific lipids in GlpG nanodiscs, and the role of the N-terminal cytoplasmic domain in these interactions. Protease activity was found to be affected by nanodisc composition, in line with previous studies on GlpG and its N-terminal truncation GlpG^{ΔN}.

Although this is an interesting study, there are significant aspects of this work that undermine the validity of some of the conclusions that have been made. These are outlined in the following points:

1) Sample purity and protease assay interpretation

The purity of GlpG samples does not seem to be very high, which can complicate interpretation of protease assay results. Gel analysis of purified GlpG (Fig. S1) shows many contaminants, and purity levels that I would estimate to be 50% or less for the GlpG samples prepared in native membranes.

We agree that the sample purity is non-optimal for the DIBMA-solubilised samples. The new protocol with a slight modification to the microfluidising step and protease inhibitor addition seems to reduce the contaminants (please see Figure 2 of this text, as well as SI Fig 1). In addition to this, we have checked the identities of the contaminating bands, and found no other active proteases that could affect activity measurements. We hope this improvement is sufficient.

While it is not surprising that GlpG in lipid nanodiscs from native membranes would co-purify with other proteins, it is misleading to state that the purity was confirmed (page 5, line 213) with no comment on the relatively low level of purity in the native membrane nanodiscs.

Interestingly, the tryptic digest performed on the repeated samples (Figure 2, this text) showed no significant integral membrane proteins co-purified with GlpG. The major contaminants were 5-methyltetrahydropteroyltriglutamate-homocysteine methyltransferase, 60kDa chaperonin, peptidoglycan associated lipoprotein, and the uncharacterized protein YjeL. We have amended the text to reflect these findings, and the increased purity is hopefully now to a satisfactory level.

Also, it is interesting that the purity of samples in Figure 4A appear to be significantly higher than those shown in Figure S1 A and B. What is the difference between these samples?

This improvement is due to the different preparation attempts – the FLAG-TatA-SUMO cleavage shown in 4A was performed on an earlier extracted GlpG F.L and ΔN. This has now been amended in the main text and SI to include only activity assays from the same batch of protease growth and purifications.

On the positive side, the purity is significantly improved after lipid exchange, which likely reflects the impact of diluting non-tagged contaminants into separate nanodiscs followed by another nickel affinity chromatography step. This seems to be a feature worth mentioning in the manuscript.

This appears true (although no integral membrane proteins were co-extracted from our analysis), the improved quality of the sample may be additionally due to the spin-concentration steps with 100 kDa MWCO post-desalt after the second purification. This would remove the contaminating proteins, as mentioned earlier they are not integral or co-extracted membrane proteins, and therefore are not

incorporated into the >100kDa DIBMALPs, so whilst this is a beneficial note for the paper, it is sadly not due to the methodology presented in this work.

Related to sample purity is a solid-state NMR spectrum (Fig. S2) that was offered as evidence for sample homogeneity. The spectrum shows the presence of folded protein, however the gels show a number of proteins are present in the sample, some at significant concentrations. With no comparison to previously published spectra it is not possible to know if GlpG alone is the source of peaks observed in this spectrum, particularly if contaminants have more favorable relaxation properties than GlpG.

This has now been amended with a comparable sample of GlpG F.L extracted in DM and reconstituted in LUVs, using the same protocol as previously published¹. As can be seen in Fig S2, the spectra are congruent with the same fold and purity in both extraction methods.

The presence of contaminating proteins in the purified nanodiscs can affect assays of protease function should any of these contaminants also have protease activity. In particular, release of the fluorescent tag from TatA-FITC could result from the action of contaminating proteases cleaving aqueous-exposed residues. It is encouraging that rhomboid-specific activity is strongly corroborated by MALDI-MS showing that at least one of the cleavage products corresponds to that expected for TatA-FITC cleaved by GlpG. However, the release of fluorescence may happen at a faster rate than GlpG-mediated cleavage, a possibility that can't be ruled out without a negative control (ideally the same experiment run on an active site serine knockout mutant).

We agree this was a failure on our part to not include this negative control – which has now been added. As can be seen in Fig. 4, and Fig. S3 and S4, the S201A activity is negligible at best, indicating that no co-extracted or contaminating proteases are present during the activity assays performed. It is worth highlighting here that the slight activity noted for the S201A mutants in the TatA-FITC assay is due to the relative dilution of the TatA-FITC embedded in the DIBMA nanodiscs into the nanodiscs containing S201A GlpG. This has the perceived effect of a slight increase in fluorescence, but is not caused by protease cleavage. This was additionally confirmed through measurement of TatA-FITC containing DIBMA nanodiscs incubated with DMPC or POPC-containing DIBMA nanodiscs, where the fluorescence also increased proportional to the quantity of lipid-only nanodiscs present (not shown). Additionally, MALDI-TOF was performed on S201A containing wells post activity measurement and showed no detectable cleavage of the TatA-FITC peptide (Fig. S5, S6).

Non-GlpG protease activity could explain the difference in rates observed between the fluorescence-based assay (minute time scale) and the gel-based assay (days). The presence of other proteases could also account for the disappearance of a contaminant band in the gel assay seen around 30 kDa in the FLAG-TatA-SUMO sample following incubation with GlpG (Fig. 4A).

This discrepancy has since been found to be due to the different preparations between Fig. 4A (gel-based assay) and TatA-FITC. As can be seen, full cleavage of TatA-FITC and SUMO-TatA-FLAG is not observed after 24 hours of incubation (TatA-FITC), nor 48 hours (in-gel assay), so the fact that cleavage is slower in DIBMA nanodiscs still holds true, however this difference is far smaller than originally estimated. The discrepancy between the original activity in the original draft may be due to either contaminations in the original preparation of the SUMO-TatA-FLAG activity assay, or due to a miscalculation in GlpG concentration due to contaminants, most likely a combination of both.

In addition, there is a difference in cleavage products between the two GlpG samples in this gel analysis, with a significant band around 13 kDa present in the GlpG cleavage not seen in the GlpG^{??}N reaction. Differences in copurification of contaminating proteases could also explain this. Taken together, all of these discrepancies raise significant questions about the activity being measured in these assays ??? how much is GlpG-specific? The use of a negative GlpG control in all of these experiments, with the active site serine knocked out, is required to establish that the measured activity was due to GlpG, and not co-purified contaminants.

We agree that the original Figure in 4A was not ideal, and do not have a reasonable explanation for the discrepancies highlighted by the reviewer. Upon repetition (and triplicate measurement/optimization) we believe we have solved these discrepancies to a satisfactory level, and with the addition of an inactive mutant for negative control. (Fig. S3)

2) Difficulties with comparisons of lipid compositions to monitor protein-lipid interactions

Lipid compositions were analyzed to investigate differences between lipid interactions with GlpG and GlpG Δ N. These two proteins were expressed in two different strains of *E. coli*, each having a different lipid composition in the membrane extracts. Although the authors suggest that the differences in lipid composition observed between nanodiscs with GlpG versus GlpG Δ N are due to differences in protein interactions with lipids, it is difficult to deconvolute the influence of GlpG on these compositions from the differences in the native membrane compositions that each sample was extracted from. It makes more sense to this reviewer to compare the composition of lipids between samples all prepared from the same strain.

We thank the reviewer for their helpful comments, and agree that this comparison required further experimentation. Initially, the strains selected were chosen from historical stocks due to the overexpression amounts required for both GlpG F.L and GlpG Δ N required for NMR, and no published data for differences in lipid composition between the strains could be found. Since submission we have attempted to obtain both versions of the protein in the same *E. coli* strain, which sadly has failed with our plasmid (please see Figure 1 and 2 of this text, as described in general comments above). We have instead been able to express the proteins in more closely related strains (C41 for full length, and C43 for Δ N) in place of the more distantly related Tuner strain. Similarly, we have measured the lipid composition of whole-cell extracts for both strains, both for non-induced and induced expression. We believe this is a sufficient control for the conclusions drawn for the amended paper. We are also careful to not directly draw comparison between the two strains in the new paper, focusing more on proof of methodology than biological findings.

However, by this analysis a different set of conclusions could be drawn in place of the conclusions provided in the manuscript. For example, GlpG in DDM or DIBMA showed enrichment of cardiolipin relative to levels observed in lipid extracts. This would appear to occur at the expense of PG and PE, a trend that was observed for both full length and truncated GlpG. One might conclude that there are specific interactions between GlpG and CL, although as the authors note, this also reflects differences in tendency for these lipids to be incorporated into DDM micelles or DIBMA nanoparticles. Nonetheless, this conclusion is different from the conclusion presented in the manuscript based on a comparison of lipid compositions of nanodiscs with GlpG from C41 versus GlpG Δ N from Tuner *E. coli* suggesting a specific interaction with PE. These different possible conclusions highlight the complications that arise when multiple variables are changed simultaneously. More reliable conclusions could be made if the analysis was done using both GlpG and GlpG Δ N expressed in the same *E. coli* strain, or with DIBMA nanodiscs prepared from the two *E. coli* strains without expressed GlpG. (I am assuming that the figure legend for figures 2A and 2D is not correct. Based on what was written in the discussion it seems like these are lipid compositions of membrane extracts that have not been solubilized by DIBMA. Whatever these samples are, the legend needs to accurately describe them, and the preparation of these important reference samples should be described in the Materials and Methods.)

We hope that the new paper draft addresses these concerns, which has slightly different findings for specific lipid retention than the original.

3) Experimental uncertainty and significant differences

There is no indication of experimental uncertainty for any of the percent lipid composition measurements, which makes it impossible to evaluate the significance of differences between samples.

The original lipid analysis is done with internal standards so whilst accurate, provides no experimental uncertainty. With the new draft biological repeats were performed for this value. This has since been amended in the main text and in SI Table 1 – we apologise for the confusion.

This is an issue throughout the analysis, for example, on page 6, line 294, a conclusion about the native lipid PE 33:1 is made that ???This retention indicates a stronger, or preferential, binding interaction than normal for a protein with bulk membrane???. Yet it is not clear what levels would be expected for these species if there was no binding interaction. If the levels of retained PE 33:1 are deemed to be above background levels, then by the same metric it seems like the amount of cardiolipin contained in these

lipid-diluted nanoparticles would also be significant, yet this was not identified. The absence of experimental uncertainty prevents objective assignment of significant versus insignificant differences, making it difficult to understand the rationale for the conclusions being presented.

We apologise for this confusion, and with the updated findings included controls for the membrane (both pre- and post- induction), and compare directly to the whole-cell extracted controls in our analysis. In addition, due to the biological duplicate (for DM-solubilised) and triplicate (for DIBMA-solubilised) we have now included error bars for more certainty in our conclusions (such as in Fig. 3, or SI Table 1 and 2).

4) Cardiolipin levels

One point of curiosity is the levels of cardiolipin being detected in this study. These are very high compared to the values that are previously seen in *E. coli* membranes expressing GlpG i.e. <5% (Reading et al, Angew Chemie Int Ed 2017). Why is there such a large amount of cardiolipin in these samples?

As pointed out by the reviewer, the reported cardiolipin content in the total *E. coli* membrane was ~5% in most cases, which is significantly lower than in the previous version of this manuscript. Therefore, we first verified the quantitative accuracy of the internal standard used in the measurement using freshly obtained standards from Avanti Polar Lipids. The analysis raised the possibility that the measured amount of CL could be overestimated by 10-15% due to the degradation/oxidation of the non-natural CLs in the internal standard. Therefore, a quantitative analysis of phospholipids was performed for the samples from new experiments in all freshly obtained internal standards (Fig. 2).

The CL content in the uninduced BL21 (C43) in the present measurement was 5.3 mol % of total phospholipid, which corresponds to the value reported in previous publications. However, the CL content in BL21 (C41) was again 11.3 mol%, which is in agreement with the value in the previous version of the manuscript. In addition, we were able to detect significantly higher levels of CL in *E. coli* cells overexpressing GlpG.

In our opinion, there could be two main reasons for the high CL content. Firstly, it has been shown that the phospholipid content of *E. coli* varies with growth conditions. Related to the conditions used in our study, it has been reported that CL content is increased upon overexpression of a membrane-associated protein² and in stationary phase³. The average OD600 of the cultures after induction of GlpG or GlpG ΔN was high due to the overnight growth at 25°C. Therefore, it is not surprising that these *E. coli* cells accumulate CL in their membranes upon induction of GlpG. It can also be assumed that the C41/C43 strains tend to have a higher CL content in order to obtain resistance to overexpression of exogenous proteins, since they are selected for this purpose.

The second reason could be technical. We quantify CLs by nano-electrospray ionisation mass spectrometry, the so-called shotgun method. This has an advantage in quantitative accuracy over the conventional LC-MS-based method. CLs are quite large and difficult to ionise, especially those with long acyl chains. We use four CL standards (from 56 to 80) and draw a standard curve to normalise the ionisation efficiency over size. As the internal standards are analysed simultaneously with the CLs from the samples, the loss due to ion suppression effect is also normalised. Please refer to our previous method reports for details (Reference 25 in the main text). In most of the measurements done in the literature, these normalisation measures weren't performed. As all methods based on mass spectrometry have drawbacks in quantitative performance, it is therefore difficult to say that our value is really correct, but we believe that most of the CL quantities measured by LC-MS in the literature have a tendency to be underestimated.

Reviewer #2 (Remarks to the Author):

In general, the experiments, results (including legends) and their interpretation are sometimes explained incomprehensibly or incompletely.

We thank the reviewer for the feedback, and apologise for the prior confusion. We hope the revised version has more (and clearer) in-depth explanations.

How often were the experiments shown in Fig 2 and 3 performed and were different protein preparations used for this purpose? There is no indication that the experiments were performed more than once.

The initially shown experiments were recorded once (with robust internal standard use for lipidomics, and therefore precise measurements can be obtained), however they were only performed on single replicates. This has now been amended to be performed on three biological replicates (DIBMA-solubilized GlpG samples), as well as two biological replicates for DM-solubilized samples. The figure texts have also been amended to include this information, where relevant.

Regarding the quality of protein preparations: I might have missed it but what is the difference between the protein preparations shown in Figure 4A compared to those in Fig S1 and S2. Figure S1, which is intended to demonstrate the purity of the proteins used, shows a contaminant in DLpG deltaN running at the height of the full length protein. Western blots using anti-his antibody with GlpG and GlpG deltaN on one gel should be added.

The initially shown Fig. 4A, S1 and S2 we performed at different times, with different preparations or batches of GlpG. We have since repeated the experiments using the same batches of numbered growths, including performing trypsin digest of gel bands (Fig. 2 of this text) to identify contaminants to rule out co-extracted proteases other than GlpG. We hope this is sufficient over the recommended anti-His western blots.

The authors state in lines 213ff that for GlpG sample homogeneity was confirmed by NMR. Here the contamination (if there?) in the GlpG deltaN preparation should be mentioned. Intact mass and tryptic digest data should be added. Protein quality and purity is an important point when it comes to interpretation of results obtained for co-purified lipids and enzyme activities.

The contamination was not detectable through NMR, and so was not mentioned in the main body of text. We have since updated the procedure, including tryptic digest information (Fig. S1) and the main text has been amended to reflect this.

Reviewer #3 (Remarks to the Author):

The paper by Sawczyc et al reported the use of DIBMALP to probe the ???native environment??? of GlpG and its activity by means of ???collisional mixing???. This paper takes for granted many unsubstantiated claims in the SMALP literatures and tries to explain its observations by aligning with some of the previous claims. In my opinion, this is a missed opportunity as the authors are better off building hypothesis based on their own observations and testing it, which will truly enrich our understanding of DIBMALP (or SMALP in general). I would like to raise a few questions for the authors to consider:

(1) The authors refer ???lipid-polymer nanoparticles??? as SMA or DIBMA. This is incorrect because SMA or DIBMA refers to the amphiphilic copolymer frequently used to produce ???lipid-polymer nanoparticles???, not the ???lipid-polymer nanoparticles??? themselves. The ???lipid-polymer nanoparticles??? should be called ???SMALP??? or ???DIBMALP??? if we follow the nomenclature proposed by Overduin and colleagues.

The use of DIBMA in the manner stated is only used in the text to specify the type of extraction method used, similar to how DDM is used to signify protein solubilized with DM, or "POPC" for POPC exchanged nanodiscs. We have amended the text to be consistent where possible with the SMALP/DIBMALP terminology however.

(2) Related to (1), the authors refer the ???lipid-polymer nanoparticles??? produced by solubilizing membranes with DIBMA copolymers as ???DIBMA nanodiscs???. This is a misleading statement frequently (unfortunately!) used by many researchers in the SMALP (or DIBMALP) community. SMALPs (or DIBMALPs) are ???lipid-polymer nanoparticles??? that cannot be simply regarded as ???nanodiscs???. In fact, it has never been explicitly proved that SMALPs (or DIBMALPs) products are all ???nanodiscs???. I would like to point to the authors recent studies using SAXS (doi.org/10.1021/acs.langmuir.1c00304) and a set of comprehensive analytical tools (doi.org/10.1021/acs.biomac.3c00034) that clearly revealed the structural heterogeneity in SMALPs (or DIBMALPs) that debunked the wrongful simplification of ???SMA

nanodiscs??? or ???DIBMA nanodiscs???. I would recommend the authors to stop using ???DIBMA nanodiscs??? as this wrongful concept misleads the audience.

The use of the term nanodiscs to describe the product formed upon the addition of an amphiphilic polymer such as SMA or DIBMA to a lipid dispersion is well established, and follows the nomenclature of the field. Whilst it is understood that the resulting nanoparticles of the crude suspension are not all nanodiscs, it is known and has been observed that the majority of the produced solubilised material is in a well-defined size definition (for DIBMA nanoparticles, this is cited as having a hydrodynamic diameter of around 18nm^{4,5}. Additionally, the vast majority of the population of nanoparticles are nanodiscs, particularly after further purification, either through centrifugation, Ni-NTA purification, or size-exclusion chromatography⁴⁻⁶, which are all commonly used methods in the literature. For example, in this work and in previous work within this lab and by the authors, the Ni-NTA purification and desalting step removes the vast majority of non-discoidal nanoparticles. The literature kindly provided by the reviewer do not perform these secondary purification steps, or even a cursory ultra-centrifugation to remove unsolubilised membrane which can explain why there is such a mixed population present.

(3) The authors repeatedly made statement about ???DIBMA nanodiscs with native lipid environment???. To the authors defense, the claim that SMALPs (or DIBMALPs) are ???native nanodiscs??? have been frequently reported in SMALPs (or DIBMALPs) literatures. This is unfortunately another bogus hypothesis in the SMALPs (or DIBMALPs) field that has never been truly substantiated. We need be serious when we make claims about ???native lipid environment???. To me, ???native lipid environment??? means both the lipid composition and organization in SMALPs (or DIBMALPs) are not altered in comparison with their parental membranes, neither of which could not be more wrong from reality. Unless proved true, the wishful thinking that SMA (or DIBMA) acts as ???cookie cutter??? to dissect native membrane without altering it has greatly misled the community and needs to be resolutely refuted in future publications. For example, lipidomic analysis (ref 27) clearly showed that that lipid compositions in SMALPs (or DIBMALPs) differ from the native membrane and from each other, and recent study (doi.org/10.1021/acs.biomac.3c00034) clearly showed that intercalation of SMA alters lipid-lipid and lipid-protein interactions. The authors need not look further but their own lipidomic analysis in Figure 2 ??? does it truly support the ???native lipid environment??? of DIBMALP? By the way, many studies made claims about ???native lipid environment??? of SMALP (or DIBMALP) based on the detection of a few endogenous lipids in SMALPs (or DIBMALPs). If we follow this logic, we might as well call detergent-solubilized membranes ???native micelles??? as they too contain endogenous lipids. In fact, the authors own lipidomic analysis (Figure 2) clearly indicates the co-extracted lipids by DDM resemble more closely to the composition of lipids in native membranes than those extracted by DIBMA. I would recommend the authors to stop repeating the wrongful claim of ???DIBMA nanodiscs with native lipid environment??? as it grossly misleads the audience.

The authors are aware of the difficulties in the field of reporting polymer-solubilised nanoparticles as native⁵, and indeed are aware of the shift away from the “cookie-cutter” description in the field in light of recent data of the past 5 years. However, the use of native lipid environment in the context of the paper is used a) within literature precedent for this sort of comparison when talking about lipid composition^{4,6-8} and b) used in the context that is explored in the paper (i.e. lipidomic analysis). The main text of the paper also goes into the potential favouring of DIBMA and associated polymers for certain lipid types that could influence the ‘native environment’ extracted by the polymer and encompassed in the nanodiscs. Therefore, the authors stand by the use of this term in this work. In regards to how polymer extraction can alter the lipid content of the extracted nanodisc, particularly when compared to DM or other detergent extracted methods, this is explored briefly in the paper text. Additional controls have since been added to address this and other remarks by the reviewers, and we hope that this is satisfactory.

(4) I would recommend the authors to be more cautious when make claim that detergent solubilization would delipdate the ???native lipids??? associated with GlpG while DIBMA would not (which is by the way another popular overly-simplified claim frequently seen in literatures). Detergents comes in many different flavors with different detergency, so does the DIBMA chains (note DIBMA is a mixture of polymer chains with different hydrophobic/hydrophilic repeating unit ratio, sequence, and molecular weight). In fact, the authors own data (Figure 2 & 4) clearly indicate DDM-solubilized GlpG doesn't delipdate GlpG.

We thank the reviewer for the recommendation, but respectfully disagree. We believe we have been sufficiently cautious when discussing the limitations of detergent extraction, with regards to the removal of bulk lipid during membrane protein extraction. However, we have amended the term to 'partially' delipidated to err on the side of more caution, and avoid confusion.

(5) Another common misconception is about the somewhat magic power of native lipids on activating membrane proteins. While the important role of native lipids on the activation of some membrane proteins have been unambiguously shown, membrane proteins are such a diverse family and we should also recognize the fact that many membrane proteins are perfectly active in non-native lipid environment, like the authors observation of increased GlpG activity in POPC or DMPC membranes. The authors tried to explain the GlpG activity by proposing that ???DDM-solubilization not leading to total delipidation, causing some native lipids from *E. coli* being transferred to the POPC or DMPC LUVs during protein reconstitution???. Not only is this claim unsubstantiated (no experiment is presented to test this claim), it is also a missed opportunity to really understand what it takes to activate GlpG.

The reviewer is correct in stating that it is a complicated field in knowing exactly the extent of the protein-lipid interaction has on all membrane proteins, and this is also true in GlpG – in fact we believe the strength of this paper is in showing a technique that could help elucidate such relationships.

The comments made in the work which the reviewer has issue with are made in reference to the cited paper in the same text⁹, which show that cell-free expressed GlpG in DMPC and POPC show a stronger activation than the samples produced in this work, and the reason put forward by the authors is due to the retained lipids in the detergent extraction (which is shown by the lipidomic analysis of this work, as well as the referenced paper). In terms of specific activation mechanisms of GlpG we believe this is beyond the scope of this paper, which focuses more on using GlpG with a known strange lipid-interaction and using it as a test case for the methodology of collisional mixing with membrane protein-containing DIBMALPs.

(6) The most important contribution of this work is the discovery that GlpG solubilized in DIBMALPs shows protease activity to its substrate, but this could only happen when GlpG and its substrate are co-localized. There could be many different hypothesis to explain why this happens at all, and the authors proposed ???collision cargo exchange??? as the mechanism following Keller???'s earlier proposal of the ???collision lipid exchange??? model, with the exception that no evidence was provided to support the ???collision cargo exchange??? Mechanism.

The evidence is provided with the extensive publications and mathematical definition by Keller *et al.* about the collisional lipid mixing between polymer-bounded nanodiscs^{10,11}, showing that the majority of the mixing between two populations of nanodiscs is through collision, not monomeric diffusion. This was further shown by the first authors DPhil thesis (<https://ora.ox.ac.uk/objects/uuid:601ebb0a-fd5e-4098-8d83-49ecc0124c17>).

Additionally, the cleavage/activity shown within the text can only be achieved through interaction of the protease and the substrate, both of which were incorporated into DIBMA nanodiscs independently, and both of which are integral membrane proteins. The only exception to this is the known soluble fluorescent probe of serine proteases, FP-TAMRA. The interaction and subsequent cleavage can only be achieved through inter-membrane interactions (i.e. protease and substrate within the same membrane). It is highly unlikely that the substrate nor the protease diffuses monomerically through the solution between the discs (as Keller and others have shown to happen on a slow scale for monomeric lipids). Additionally, the TatA substrate is cleaved in a highly specific location (as shown by the MALDI-TOF spectra), which can only occur if both GlpG and TatA are within the same membrane when meeting. Therefore, the only possible explanation is the mixing of both lipid and protein between nanodiscs, i.e. collisional lipid mixing as proposed by Keller *et al.*

The authors also did not make it clear what is the ???cargo??? in the exchange? The lipids, the substrate, or the GlpG ?

We apologise for the confusion, it is both lipid and protein (both substrate and GlpG) being exchanged. We proposed the use of this term as “collisional lipid mixing” as defined by Keller *et al.* refers only to the lipid cargo in the DIBMALPs being exchanged, but due to the cleavage shown by our work it is

clear that integral membrane proteins are also exchanged. We have amended the originally suggested 'collisional cargo mixing' to just 'collisional mixing' to avoid confusion over the identity of the cargo.

By the way, lipid exchange among DIBMALPs or SMALPs or detergent micelles in general occurs naturally through the dynamic equilibrium of lipids in the self-assembled nanostructures with that freely dissolved in solution, and this is nothing new. Keller's earlier contribution (ref 18) is that they established kinetic models to describe the two routes (i.e., freely diffusion vs collision exchange) and measured the exchange rates to show that different exchange rates exist, and the exchange rates are dependent on ionic strength and SMALP sizes (doi.org/10.1016/j.eurpolymj.2018.09.043). This observation indirectly supports the possibility of a collision exchange model, or as Keller puts it, the collisional exchange dominates at millimolar lipid concentrations, whereas diffusional transfer dominates at lower concentrations.

We thank the reviewer for the reminder about monomeric diffusion for SMALPs, DIBMALPs (and even LUVs, peptide nanodiscs etc). We respectfully disagree that this is indirectly supported. The publications referred to by the reviewer and in our work clearly demonstrate (and mathematically describe in the case of Keller's work) that the primary mixing method in the concentrations of nanodiscs used in this work are through collisional lipid mixing, not through singular lipid diffusion alone.

This is an interesting hypothesis rather than a slam dunk conclusion that everyone would agree. In this context, the authors need make clear what collision cargo exchange means in their hypothesis, and design experiment to test their hypothesis. Without that data, I can't recommend proposing of this new concept as it is no more than another scientific fantasy.

We thank the reviewer for their input, however respectfully point out that the collisional mixing (whilst perhaps ill-defined in terminology in the initial text but we hope has since been clarified satisfactorily) was thoroughly tested in the work, both by cleavage assays between GlpG and SUMO-TatA-FLAG (which can only occur through such mixing), the rapid exchange of lipids of GlpG-DIBMALPs and DMPC or POPC in DIBMA nanodiscs (which kinetically is plausible only through collisional lipid mixing as described by Keller et al.). In addition, the cleavage of TatA-FITC which was shown by MALDI-TOF to cleave exactly where expected if both protease and substrate are within the same membrane - which is not possible through the single diffusion of lipids hypothesis. We believe that the combination of all of these data points, including the lipidomic analysis is a strong argument for this methodology.

Reviewer #4 (Remarks to the Author):

Studies have shown the importances of avoiding detergents and using near-native membrane mimetics such as nanodiscs in the functional reconstitution of membrane proteins and further structural studies. In particular, synthetic polymers are increasingly used to develop nanodiscs with various size and stability under different conditions including pH, T, and the presence of divalent metal ions. One of the unique properties of these nanodiscs is their ability to collide and exchange lipid contents. The rate of collision, stability, and stability-size-homogeneity have been shown to vary with the type of belt (and charge) and the lipid composition. In this study, the authors well utilized this unique property to demonstrate the lipid environment within the DIBMA-based nanodiscs can be exchanged with non-native lipids. In addition, this property is effectively used to explore the different effects non-native lipids have on the activity of GlpG and GlpG^N. These are exciting results and will be useful to researchers in the field. The manuscript is well written. Therefore, I strongly recommend the publication of the manuscript after revision.

1) While lipid-exchange is an useful property of the polymer or peptide based nanodiscs, it can be a limitation if size homogeneity and stability are desired. This should be mentioned.

We have amended the text to include this addition within the discussion:
"This collisional mixing method in polymer-based nanodiscs (SMALPs/DIBMALPs) is an ideal simple methodology for minimally invasive lipid exchange and the exploration of lipid-protein association. Similar techniques, such as those in peptide-based nanodiscs have been also shown also. These nanodiscs possess increased homogeneity over polymer systems, but require detergent for their production."

2) The authors should briefly mention and include the studies that demonstrated (using high-speed AFM and ³¹P NMR in real-time) the lipid-exchange process (Chem. Mater. 2018, 30, 10, 3204) and its use to determine the type of lipids preferred by membrane-bound cytochrome proteins (Angew Chem Int Ed Engl. 2018, 57(13):3391; Chem.Comm. 2018, 54(49):6336).

We have added to the introduction the following section and thank the reviewer for pointing them out:

“Similarly, peptide-based nanodiscs have shown some propensity for lipid mixing, either through single lipid diffusion¹², or through nanodisc fusion^{13–15}, however they are still limited with detergent-based formation methods.”

3) A recent study based on non-ionic (inulin based) polymer demonstrated that the nanodiscs collision and fusion can result in size heterogeneity. But, it can be overcome by tuning the charge of the lipids encased.

We thank the reviewer for the recommendation, which was read with interest when published. In particular the look into the difference between non-ionic and ionic (or just ‘charged’) polymer nanodiscs. Whilst size heterogeneity was not the focus of this study (although the previously mentioned references in (2) have since been added with amendments), it is an interesting question (and has been preliminarily been observed by us through TEM and cryo-EM samples) on the overall stability and heterogeneity of such polymer nanodiscs in protein structural biology. However we believe this exact reference, with the focus on non-ionic and ionic polymer belts and lipid-only nanodiscs, is beyond the purview of the current manuscript as submitted.

1. Shi, C. *et al.* Structure and Dynamics of the Rhomboid Protease GlpG in Liposomes Studied by Solid-State NMR. *J. Am. Chem. Soc.* **141**, 17314–17321 (2019).
2. Carranza, G. *et al.* Cardiolipin plays an essential role in the formation of intracellular membranes in *Escherichia coli*. *Biochim. Biophys. Acta - Biomembr.* **1859**, 1124–1132 (2017).
3. Cronan, J. E. Phospholipid Alterations During Growth of *Escherichia coli*. *J. Bacteriol.* **95**, 2054–2061 (1968).
4. Oluwole, A. O. *et al.* Solubilization of Membrane Proteins into Functional Lipid-Bilayer Nanodiscs Using a Diisobutylene/Maleic Acid Copolymer. *Angew. Chemie - Int. Ed.* **56**, 1919–1924 (2017).
5. Sawczyc, H., Heit, S. & Watts, A. A comparative characterisation of commercially available lipid-polymer nanoparticles formed from model membranes. *Eur. Biophys. J.* (2023) doi:10.1007/s00249-023-01632-5.
6. Orwick-Rydmark, M. *et al.* Detergent-free incorporation of a seven-transmembrane receptor protein into nanosized bilayer lipodisc particles for functional and biophysical studies. *Nano Lett.* **12**, 4687–4692 (2012).
7. Reading, E. *et al.* Interrogating Membrane Protein Conformational Dynamics within Native Lipid Compositions. *Angew. Chemie - Int. Ed.* **56**, 15654–15657 (2017).
8. Bada Juarez, J. F., Harper, A. J., Judge, P. J., Tonge, S. R. & Watts, A. From polymer chemistry to structural biology: The development of SMA and related amphipathic polymers for membrane protein extraction and solubilisation. *Chem. Phys. Lipids* **221**, 167–175 (2019).
9. Engberg, O. *et al.* Rhomboid-catalyzed intramembrane proteolysis requires hydrophobic matching with the surrounding lipid bilayer. *Sci. Adv.* **8**, 1–10 (2022).
10. Cuevas Arenas, R. *et al.* Fast Collisional Lipid Transfer among Polymer-Bounded Nanodiscs. *Sci. Rep.* **7**, 1–8 (2017).
11. Grethen, A., Glueck, D. & Keller, S. Role of Coulombic Repulsion in Collisional Lipid Transfer Among SMA(2:1)-Bounded Nanodiscs. *J. Membr. Biol.* **251**, 443–451 (2018).

12. Zhang, G. *et al.* Identifying Membrane Protein–Lipid Interactions with Lipidomic Lipid Exchange-Mass Spectrometry. *J. Am. Chem. Soc.* (2023) doi:10.1021/jacs.3c05883.
13. Ravula, T. *et al.* Real-Time Monitoring of Lipid Exchange via Fusion of Peptide Based Lipid-Nanodiscs. *Chem. Mater.* **30**, 3204–3207 (2018).
14. Barnaba, C. *et al.* Lipid-exchange in nanodiscs discloses membrane boundaries of cytochrome-P450 reductase. *Chem. Commun.* **54**, 6336–6339 (2018).
15. Barnaba, C. *et al.* Cytochrome-P450-Induced Ordering of Microsomal Membranes Modulates Affinity for Drugs. *Angew. Chemie Int. Ed.* **57**, 3391–3395 (2018).

REVIEWER COMMENTS

Reviewer #1 (Remarks to the Author):

The significantly modified manuscript by Sawczyk et al uses lipidomics and intramembrane protease activity to demonstrate that mixing between DIBMA-solubilized samples can be used to allow exchange both protein and lipid components between complexes. The significance of this work will be high for groups that wish to study membrane protein function involving transmembrane substrates in native lipid environments without having to expose their samples to detergents. While there were several concerns with the previous version of the manuscript, significant improvements have been implemented that better support the more conservative conclusions that have been made in the revised submission.

In their rebuttal letter, the authors provide a thorough account of the efforts and challenges associated with expression of GlpG, justifying the necessity of using different strains for the full-length and truncated constructs. Lipidomics analyses were also done for more similar *E. coli* strains that are more appropriate to evaluate the impact of GlpG expression, particularly with the addition of pre- and post-induction samples to the analysis. The inclusion of biological replicates for lipidomics experiments is also a helpful addition. The large amount of speculative identification of specific protein-lipid interactions has been removed, and replaced by the statement that there is no evidence suggesting specific retention of any particular lipid type by GlpG, a reasonable conclusion that is consistent with the available data.

A change has been implemented in the purification of GlpG in DIBMA that has improved the purity compared to that from the previous submission. Inconsistencies in purity levels that were previously an issue have been reduced by application of the same improved purification protocol to all samples analyzed in the updated work. That being said, the purity of the samples should be estimated instead of stating that the purity is 'confirmed' (Line 265) since this is needed to provide an informative assessment of the degree of the purification. Also, there is still some variability from preparation to preparation which in itself is not unusual but may be responsible for the varying degrees of degradation of purified detergent-solubilized GlpG samples seen in Figure S1 panels C and D. This is likely the cause for the variability in activity between biological replicates seen in Figure 4A, since the second replicate showed significant amounts of degradation. My recommendation would be to remove the activity data from this sample. As it is the detergent-solubilized control repeating results from previous studies, it is not necessary to show the additional dataset to demonstrate that the assays are working. Fortunately the proteolytic degradation of GlpG did not seem to be an issue with the DIBMA-solubilized samples.

Another significant improvement is the inclusion of negative controls for protease assays using the active-site serine knockout. This provides better support for the conclusions that rhomboid protease activity is being observed, with minimal contributions from contaminating proteases. However the data is less convincing for the gel-based activity assays of DIBMA-solubilized GlpG using SUMO-Tat-FLAG as the substrate. While the assay worked well on samples purified from detergent, in DIBMA-solubilized samples the bands at the molecular weight suggested to be that of the cleavage product are very weak and differences in band intensities between the zero time point and the post-incubation time point are difficult to see for many samples. In addition, judging from the decrease in intensity of the uncleaved substrate in the series with truncated GlpG, there appears to be similar amounts of GlpG degradation in all samples including that of the negative control S201A. As the purpose of this assay is to demonstrate exchange of protein between DIBMA-solubilized complexes, it needs to be of higher quality to support this conclusion. Nonetheless, the FITC-Tat based assays do give good evidence that GlpG is cleaving the transmembrane substrate which provides support for the transfer of proteins between DIBMA-solubilized complexes.

Some other minor issues that should be fixed in the paper are listed here:

- The figure legend for Figure S1 makes a reference to Figure S2 that does not match its description.
- The description for S1D suggests that the samples are detergent solubilized in various lipid environments. It should be clarified that they were reconstituted from DM into liposomes.
- Figure S1C has regions of the band that were analyzed by MS. According to the figure legend, the region that labelled 2* is not GlpG. However, there are 2 bands encapsulated by this region. Which one was submitted for MS analysis? Presumably one of these bands must be the truncated GlpG?
- Line 219: It is stated that “GlpG and GlpGAN extracted into DIBMALPs have approximately 6× the lipid:protein ratio of GlpG and GlpGDN extracted into DM micelles.” How was this determined? This information should be included in the Materials and Methods.
- Line 251 – The reference to Fig. S3 D should be changed to Figure S1 D.
- Line 254 – There is a statement that MALDI-TOF confirmed GlpG-specific cleavage should be softened as there were some experiments where MALDI profiles only sometimes showed this evidence of specific cleavage. For example, lipid-exchanged DIBMA GlpG showed barely any intensity for the expected cleavage product (Fig S6 C, I, O) or very low intensity in spite of high activity as measured by fluorescence (Fig S6 B, E, H). This was also the case for liposome-reconstituted GlpG (Fig S5 H, I, M, O, P, Q), which in some cases also showed the presence of degradation products that were smaller than expected. Variability in the Tat-FITC results between biological trials was also significant. Aside from these exceptions, expected cleavage patterns were largely consistent with predictions which does suggest that specific GlpG-mediated cleavage was responsible for the observed proteolysis.
- Line 261 – The statement of a significant decrease in activity for truncated GlpG purified in DM and reconstituted into non-native lipids was only true for the Tat-FITC experiments. It should be

acknowledged that the gel-based assay with SUMO-Tat-FLAG showed higher activity in PC relative to E. coli lipid samples.

- Line 284 – The secondary cleavage product was not present in Figure S6 panel O. This statement should be modified to indicate that the secondary cleavage product was detected in almost all MALDI-TOF profiles recorded with truncated GlpG in PC-washed samples.

Reviewer #2 (Remarks to the Author):

In many cases, the authors convincingly addressed the reviewers' criticisms. However, a few ambiguities remain, which should be addressed.

1) Figure 4B shows three replicates for GlpG (DIBMA1-3), where three significantly different ratios of PMPC versus POPC are obtained, which is not discussed.

What is the reason for this large discrepancy?

This somewhat calls into question the the use of this method to test the effects of lipid species on protease activity.

2) In their rebuttal letter, the authors state: 'The CL content in the uninduced BL21 (C43) in the present measurement was 5.3 mol % of total phospholipid, which corresponds to the value reported in previous publications. However, the CL content in BL21 (C41) was again 11.3 mol%, which is in agreement with the value in the previous version of the manuscript. In addition, we were able to detect significantly higher levels of CL in E. coli cells overexpressing GlpG.'

This statement seems to be a mistake. In the revised version, the CL content of BL21 (C41) is 11.3 mol%, whereas in the original manuscript it was 22.9 mol% (Figure 2A).

If the CL content of BL21 C41 is identical in the original submission and the revised version, this would contradict to the author's statement that artificially high CL values were obtained in the original submission due to a degraded/oxidized CL standard.

3) Page 2, lines 74ff

The authors state 'For detergent (n-Decyl β -maltoside (DM), Glycon) purified samples, purification followed previously described protocols.

Please add a reference.

In the original version the authors used DDM. Why was the detergent changed to DM?

4) Line 142

Please add the pmol amounts of standard mixtures added to the extraction.

Reviewer #3 (Remarks to the Author):

It has been well known that lipid exchange occurs between detergent-lipid micelles, liposomes, protein-encased nanodiscs, and SMALPs. The contribution of this study is the proposed concept that protein solubilized in DIBMALPs can also exchange among different protein-carrying DIBMALP populations via collisional mixing, which enables the study of protein-protein interactions (such as the catalytic cleavage of a protein substrate by GlpG shown in this study) under native environment of DIBMAPs (suppose the presumption that DIBMALPs represent native environment AND that the collisional mixing does not change such environment is true). The revised version added new data to mend some of the discrepancy shown in the original submission on GlpG expression and purification, activity assay, and lipidomic analysis. It does not provide new data to address the questions raised by Reviewer #3 regarding the structure of protein-carrying DIBMALPs and experimental evidence of protein migration via collisional mixing. Those are rather important questions that need to be answered with experimental data to support the authors' hypothesis. Let me explain why.

DIBMAs (or SMAs for that matter) are essentially a mixture of amphiphilic random copolymer chains of different detergency. The detergency of individual chains depends on its amphiphilic balance (i.e., DIB/MA ratio, sequence, and chain size). The earlier "cookie-cutter" model of how DIBMAs (or SMAs) solubilize membranes has been proven to be only a fantasy by recent data, and for that reason, we should use DIBMALPs instead of nanodiscs or native nanodiscs or native-like nanodiscs unless we have evidence to show that we have successfully purified a portion of DIBMALPs (or SMALPs) to retrieve such a truly-existed population, because the solubilized membranes (i.e., DIBMALPs or SMALPs) is also a mixture of polymer-lipid self-assembled particles with different degrees of polymer-lipid inter-mixing corresponding to the different detergency of individual DIBMA chains as well as the polymer-to-lipid ratios used to solubilize the membrane.

Although we all love the concept of detergent-free solubilization of membrane proteins, we need to be mindful that plenty of chains in any given DIBMA (or SMA) sample behave just like detergent because their individual detergency is strong. The use of centrifugation to remove undissolved membranes and Ni-NTA resin to separate out the protein-carrying DIBMAPs as shown in this study doesn't necessarily produce nanodiscs; those selected protein-carrying DIBMAPs could well be a mixture of DIBMA-GlpG-lipid mixed micelles. In fact, previous study shown that the average GlpG-to-lipid ratio in DIBMAPs is ~1:10 (DOI: 10.1021/jacs.8b08441), which is very similar to what observed in detergent-solubilized GlpG (~1:14; DOI: 10.1016/j.jmb.2011.01.029) but far below what should be expected from nanodiscs with a diameter of 12-15 nm, suggesting a very large portion of protein-carrying DIBMAPs are actually in micelle forms. Without separating and identifying the membrane-like DIBMAPs fractions from the micelle-like DIBMAPs fractions, the entire argument about protein hopping among different protein-carrying DIBMAP populations via collisional mixing stays as an unvalidated hypothesis because the fusion between micelles (DOI: 10.1021/jacs.7b09060) or between micelles and membranes (DOI: 10.1016/j.bpj.2021.04.012) could also explain the observed GlpG cleavage activity.

I'd like to propose the following suggestions for the authors to consider:

(1) Separate and identify the membrane-like DIBMAPs fractions.

Please show an SEC chromatogram of protein-carrying DIBMAPs. Show both dRI and UV-Vis signals. Calibrate the elution volume with protein standards of known hydrodynamic radius. At the very minimum, measure the GlpG-to-lipid ratio at a few selected elution volumes and explain which fractions will be collected for your study.

(2) The collisional mixing

At the bottom of any mixing (no matter how we call it) is the dissociation of species x from party A, diffuse of free x in the medium, and association of free x into party B. This could happen via the slow dissociation-association equilibrium of x between party A (or party B) and the free x in the medium, or the accelerated equilibrium of the same route, i.e., collisional mixing as originally coined by Nichols in the study of lipid transfer between detergent-lipid micelles (DOI: 10.1021/bi00411a006), where the collisional mixing was used to refer the accelerated rate of x dissociation from party A (or party B) - which is the rate-limiting step of the lipid mixing - due to collisions among party A and/or B. Keller et al (DOI: 10.1038/srep45875) showed that accelerated lipid transfer (i.e., collisional mixing) also happens among SMALPs when bulk solution concentrations of lipid in the donor and acceptor populations are high enough. As a side note for Keller's work, the SMA with a S/MA=3/1 ratio was used, this variant of SMA is less popular in SMALP community because it is known to have strong detergency. As such, a significant portion of SMALPs in that study is likely in polymer-lipid micelle forms, so the observed fast lipid transfer is in line with Nichols' original study of lipid transfer between detergent-lipid micelles. The current study takes a leap of faith by proposing the collisional mixing of proteins among different protein-carrying DIBMAPs without any limitations

(such as concentration or DIBMAP structure) or supporting evidence besides the observed GlpG cleavage activity. Proteins don't magically hop between DIBMALPs. This proposal basically suggests that GlpG or its protein substrate can be released from one DIBMALP into buffer in their free forms via collision, then reinserting themselves into another DIBMALP. Considering the large hydrophobic domain of both proteins and their strong interactions with the surrounding polymer-lipid matrix, this is a bold hypothesis and appears to be far less likely than a simple fusion of DIBMALPs. Without any experimental data to exclude the possibility of fusion among DIBMALPs, the observed cleavage activity is not sufficient to support the authors' hypothesis because fusions of different DIBMALPs could also bring GlpG close to its substrate.

(3) Unless I missed anything, I'm not sure how the authors concluded that GlpG and GlpGΔN extracted in DIBMALPs have 6x lipid:protein ratio than those extracted in DM (line 219-220). Please provide supporting data on that. I'm also not entirely sure what is the rationale to compare the activity between GlpG and GlpGΔN and what important discovery is drawn out of this comparison. I recommend the authors to explain the rationale of this experimental design in the abstract, and to elaborate the important discovery, if any, in the summary.

(4) There are a number of habitual talking points in the DIBMALP or SMALP community that are not entirely true, I'm listing a few examples below for the authors to reconsider:

- Title: ... to probe the native environment of...

Unless I missed anything, I don't see any evidence about the native environment enabled by DIBMALPs. The authors' own lipidomic analysis confirmed that no significant lipid composition difference is observed between the DM- and DIBMA-solubilized GlpG (Fig 2), and no specific lipid species is observed to be tightly bound to DM- and DIBMA-solubilized GlpG, or retained after DMPC or POPC washing (Fig. 3).

- Line 30-31: ... the use of detergent in protein extraction and purification leads to a loss of the native lipid environment...

Extract of proteins from native membranes by any means, SMA or DIBMA included, would lead to a loss of the native lipid environment of the proteins. Period.

- Line 36-37: ...Lipid-polymer nanodiscs, such as SMALPs and DIBMALPs, have been shown to extract integral membrane proteins along with their native lipids into nanodiscs, without the use of detergents at any step ...

Again, SMALPs and DIBMALPs are "polymer-lipid nanoparticles" (original definition) instead of "nanodiscs" because of their diverse structures; SMA and DIBMA themselves could have

detergency similar to detergents and behave just like detergents; detergents are also known to extract integral membrane proteins along with their native lipids as shown in many protein crystallography studies as well as the authors' own data (Fig 2 & 3).

- Line 38-39: ...Proteins within these discs often possess increased stability compared to their detergent solubilized counterparts...

I don't see any systematic study to validate this generalized statement. Just like SMA or DIBMA that can be made with different detergency, many different detergents exist with different detergency too. We may compare the stability of a specific polymer-solubilized proteins with a specific detergent-solubilized proteins, but a generalized statement such as this one does not make any sense.

Reviewer #5 (Remarks to the Author):

Review of "Using collisional mixing in lipid-polymer nanoparticles to probe the native environment of intra-membrane rhomboid protease GlpG and its associated activity" by H. Sawczyk et al.

Given that this paper has already undergone extensive review, I will limit my commentary to the discussion between the authors and reviewer 3.

This paper presents evidence of lipid mixing in a DIBMALP containing protein. I believe that the community has been aware of such lipid mixing for a while, in fact, it was used by Edler et al. (see <https://doi.org/10.1016/j.bbadv.2021.100033>) to exchange deuterated lipids into an OMPF containing SMALP in at least 2021. However there has been no systematic study of this process and as such this paper certainly merits publication on that basis alone. The paper also suggests that proteins can mix within different populations of DIBMALPs which allows for studies of protein-protein interactions as illustrated here. This is also a very interesting result and adds to the case for publication.

That said, reviewer 3 is making an extremely valid point that the simplistic view of these particles as nanodiscs is not really accurate (this is nicely demonstrated in the article <https://doi.org/10.1021/acs.biomac.3c00034> that was referenced by reviewer 3 previously) and that the authors are lacking some evidence to support their mechanistic conclusions, particularly on collisional mixing to exchange proteins. I agree with reviewer 3 that the authors have not provided sufficient evidence that this is the mechanism of exchange, although it is not an unreasonable hypothesis based on the literature.

Reviewer 3 is suggesting that the authors should obtain substantially more experimental evidence to support their claims. I think this is unnecessary since that will clearly add a substantial delay to publication. In my view the result that lipids and proteins can exchange between DIBMALPs (whatever their form) is already sufficiently interesting for publication and that the detail of structure and mechanism would be suitable for future publications. My suggestion to resolve the impasse between the authors and reviewer 3 would be for the authors to tone down their description and add appropriate caveats. For example, use of the term nanodisc could be qualified in a referenced footnote that explains the known variability in these particles and the cartoons shown in figs 1 & 5 could add a similar caveat. Similarly, to avoid propagating the other misconceptions within the SMALP/DIBMALP community that are highlighted by the reviewer, I suggest a careful rephrasing of the introduction. I do think that the authors should be free to make an hypothesis regarding the exchange mechanism, as long as it is clear that this is their opinion and that the evidence for structure and mechanism within their study is limited. The addition of a short discussion and a reference to other possible mechanisms such as micelle fusion as suggested by reviewer 3 would certainly not detract from the main conclusions of the paper.

Reviewer #1 (Remarks to the Author):

The significantly modified manuscript by Sawczyk et al uses lipidomics and intramembrane protease activity to demonstrate that mixing between DIBMA-solubilized samples can be used to allow exchange both protein and lipid components between complexes. The significance of this work will be high for groups that wish to study membrane protein function involving transmembrane substrates in native lipid environments without having to expose their samples to detergents. While there were several concerns with the previous version of the manuscript, significant improvements have been implemented that better support the more conservative conclusions that have been made in the revised submission.

In their rebuttal letter, the authors provide a thorough account of the efforts and challenges associated with expression of GlpG, justifying the necessity of using different strains for the full-length and truncated constructs. Lipidomics analyses were also done for more similar *E. coli* strains that are more appropriate to evaluate the impact of GlpG expression, particularly with the addition of pre- and post-induction samples to the analysis. The inclusion of biological replicates for lipidomics experiments is also a helpful addition. The large amount of speculative identification of specific protein-lipid interactions has been removed, and replaced by the statement that there is no evidence suggesting specific retention of any particular lipid type by GlpG, a reasonable conclusion that is consistent with the available data.

A change has been implemented in the purification of GlpG in DIBMA that has improved the purity compared to that from the previous submission. Inconsistencies in purity levels that were previously an issue have been reduced by application of the same improved purification protocol to all samples analyzed in the updated work. That being said, the purity of the samples should be estimated instead of stating that the purity is 'confirmed' (Line 265) since this is needed to provide an informative assessment of the degree of the purification. Also, there is still some variability from preparation to preparation which in itself is not unusual but may be responsible for the varying degrees of degradation of purified detergent-solubilized GlpG samples seen in Figure S1 panels C and D. This is likely the cause for the variability in activity between biological replicates seen in Figure 4A, since the second replicate showed significant amounts of degradation. My recommendation would be to remove the activity data from this sample. As it is the detergent-solubilized control repeating results from previous studies, it is not necessary to show the additional dataset to demonstrate that the assays are working. Fortunately the proteolytic degradation of GlpG did not seem to be an issue with the DIBMA-solubilized samples.

We once again thank the reviewer for their constructive feedback throughout this process, and have removed the second DM-solubilized GlpG and GlpG Δ N from this figure. However, to avoid too much cropping of gel images, we have retained the replicates in the SI data, as well as the raw cleavage data in SI Fig 4 including the replicates. Additionally, we have also removed the wording saying that purity was 'confirmed' and have instead said that purity was checked by SDS-PAGE.

Another significant improvement is the inclusion of negative controls for protease assays using the active-site serine knockout. This provides better support for the conclusions that rhomboid protease activity is being observed, with minimal contributions from contaminating proteases.

However the data is less convincing for the gel-based activity assays of DIBMA-solubilized GlpG using SUMO-Tat-FLAG as the substrate. While the assay worked well on samples purified from detergent, in DIBMA-solubilized samples the bands at the molecular weight suggested to be that of the cleavage product are very weak and differences in band intensities between the zero time point and the post-incubation time point are difficult to see for many samples. In addition, judging from the decrease in intensity of the uncleaved substrate in the series with truncated GlpG, there appears to be similar amounts of GlpG degradation in all samples including that of the negative control S201A. As the purpose of this assay is to demonstrate exchange of protein between DIBMA-solubilized complexes, it needs to be of higher quality to support this conclusion. Nonetheless, the FITC-Tat based assays do give good evidence that GlpG is cleaving the transmembrane substrate which provides support for the transfer of proteins between DIBMA-solubilized complexes.

We agree with the reviewer that the gel-based assay, although designed to be qualitative and complementary in nature to the TatA-FITC cleavage data, is not as high-quality as we would have liked, this is why it is in the SI and only referenced qualitatively in the text. We hope this suffices, and that its continued inclusion in this manner is palatable to the reviewer.

Some other minor issues that should be fixed in the paper are listed here:

- The figure legend for Figure S1 makes a reference to Figure S2 that does not match its description.
- The description for S1D suggests that the samples are detergent solubilized in various lipid environments. It should be clarified that they were reconstituted from DM into liposomes.
- Figure S1C has regions of the band that were analyzed by MS. According to the figure legend, the region that labelled 2* is not GlpG. However, there are 2 bands encapsulated by this region. Which one was submitted for MS analysis? Presumably one of these bands must be the truncated GlpG?
- Line 219: It is stated that “GlpG and GlpGΔN extracted into DIBMALPs have approximately 6x the lipid:protein ratio of GlpG and GlpGΔN extracted into DM micelles.” How was this determined? This information should be included in the Materials and Methods.
- Line 251 – The reference to Fig. S3 D should be changed to Figure S1 D.
- Line 254 – There is a statement that MALDI-TOF confirmed GlpG-specific cleavage should be softened as there were some experiments where MALDI profiles only sometimes showed this evidence of specific cleavage. For example, lipid-exchanged DIBMA GlpG showed barely any intensity for the expected cleavage product (Fig S6 C, I, O) or very low intensity in spite of high activity as measured by fluorescence (Fig S6 B, E, H). This was also the case for liposome-reconstituted GlpG (Fig S5 H, I, M, O, P, Q), which in some cases also showed the presence of degradation products that were smaller than expected. Variability in the Tat-FITC results between biological trials was also significant. Aside from these exceptions, expected cleavage patterns were largely consistent with predictions which does suggest that specific GlpG-mediated cleavage was responsible for the observed proteolysis.
- Line 261 – The statement of a significant decrease in activity for truncated GlpG purified in DM and reconstituted into non-native lipids was only true for the Tat-FITC experiments. It should be acknowledged that the gel-based assay with SUMO-Tat-FLAG showed higher activity in PC relative to E. coli lipid samples.
- Line 284 – The secondary cleavage product was not present in Figure S6 panel O. This statement should be modified to indicate that the secondary cleavage product was detected in

almost all MALDI-TOF profiles recorded with truncated GlpG in PC-washed samples.

We have amended the text in the manners requested above, and thank the reviewer for their keen attention to detail in identifying the mistakes.

As for the 2* band in SI Fig. 1C, the double bands referenced were both taken for MS and analysed together, with the recorded intensity being around 4x the signal intensity of the next detectable protein, the 50S ribosomal protein L13. The doubling in bands can sometimes be observed for GlpG in our hands, and we believe it is the polyphosphorylation of the His tag (mentioned in papers such as <https://pubmed.ncbi.nlm.nih.gov/9918669/>). On an anecdotal note, we've noticed this double banding more since the switch to NuPAGE gels, which we believe is due to the higher resolution of the NuPAGE gels. The truncated GlpG (i.e. GlpGΔN) has a calculated mass of 23.8kDa (compared to the 33.5kDa of wild-type GlpG), so it is highly unlikely that the two bands in the gel are the wild-type and truncated GlpG running together for this lane. However, it is a feature in the lane labelled FL 1 (1*), and faintly FL 2. Later work (not shown) has indicated that the N-terminal domain is most likely the lower bands present in the FL 1 and 2 lanes (running at approximately 12kDa).

For the lipid:protein ratio question, the value comes from the averaged DDM and DIBMA samples for the initially purified samples (excluding the S201A controls). It is also now attached in the SI as SI Table 3, for completeness for any reader in the future, and as similar questions were raised by reviewer 3. The averaged values come from the initial purification values for both GlpG and GlpGΔN, with the measurement error the protein:lipid ratio for DIBMA is around 6x higher than the detergent value.

Reviewer #2 (Remarks to the Author):

In many cases, the authors convincingly addressed the reviewers' criticisms. However, a few ambiguities remain, which should be addressed.

1) Figure 4B shows three replicates for GlpG (DIBMA1-3), where three significantly different ratios of PMPC versus POPC are obtained, which is not discussed.

What is the reason for this large discrepancy?

This somewhat calls into question the the use of this method to test the effects of lipid species on protease activity.

We thank the reviewer for their positive comments on the amount of work performed to address the previous feedback. As for the discrepancy, there are a couple possible explanations: 1) Generic variability between samples. As shown in the high errors between the DIBMA samples 1-3 for a single lipid (say DMPC), there is a large error between the amount of lipid 'washed in', where one would expect the same amount per sample. Whilst this is not an explanation in itself, this discrepancy is most likely due to the measurement of protein concentration (and therefore the relative amount of lipid used) being based upon on an inaccurate concentration (i.e. the concentration as measured by Abs₂₈₀ over more thorough and time-consuming methods such

as BCA or Bradford). This would leave to relative concentrations of protein:lipid varying between biological replicates. This could be solved by using a vast excess (i.e. over the 100x lipid excess used in this work) to reduce the effect of the protein concentration variation. 2) Differences between the kinetics of DMPC and POPC DIBMALPs. DMPC and POPC were prepared by powder weighing (DMPC) and chloroform stock measurement (POPC), which have different problems in relation to lipid quantity accuracy. Additionally, it is known that the mixing rates measured in lipid-only DIBMALPs and SMALPs vary depending on the temperature (<https://ora.ox.ac.uk/objects/uuid:601ebb0a-fd5e-4098-8d83-49ecc0124c17> , Chapter 4). Whilst this is assumed to be due to the relative kinetics of the particles, it could also be due to the elasticity of the lipids within the particles. If this is true, then the elasticity of the nanodiscs could affect the relative ratios of lipids in collision, and POPC and DMPC being different lipids with varying melting temperatures (and therefore assumedly different fluidity), this may be the cause here. This is actually the reason POPC and DMPC were tested, over just DMPC or POPC. Both options could hopefully be solved with a higher excess of non-native lipid DIBMALPs, or a longer incubation time.

It is correct also that this limits the conclusions of this paper on how non-native lipids affect protease activity, this is why the conclusions have been 'toned down' from the original draft. However, with more careful measurements of both lipid and protein concentrations, or the study of more specific lipid-protein interactions for different protein systems, we believe it has the potential to be a robust technique. The exact mixing mechanism, and the effect of different lipids, polymers, and proteins on this, are of course reasonable subjects for future more in-depth work.

2) In their rebuttal letter, the authors state: 'The CL content in the uninduced BL21 (C43) in the present measurement was 5.3 mol % of total phospholipid, which corresponds to the value reported in previous publications. However, the CL content in BL21 (C41) was again 11.3 mol%, which is in agreement with the value in the previous version of the manuscript. In addition, we were able to detect significantly higher levels of CL in *E. coli* cells overexpressing GlpG.'

This statement seems to be a mistake. In the revised version, the CL content of BL21 (C41) is 11.3 mol%, whereas in the original manuscript it was 22.9 mol% (Figure 2A).

If the CL content of BL21 C41 is identical in the original submission and the revised version, this would contradict to the author's statement that artificially high CL values were obtained in the original submission due to a degraded/oxidized CL standard.

We apologise for the confusion as there is a slight mistake in the naming of the cells used in the quoted text. For the C41 uninduced cells the CL content was 11.3%, and for C43 the uninduced cells was 5.3% (Table S1). The value of 5.3% (and later the 8.7% for the induced cells of C43) is closer to previously reported CL values for GlpG expressed in BL21 (C43) cells by Reading *et al.* (10.1002/anie.201709657) of 3.1% (+/- 1.6). However, the values obtained in the revised version of our work were consistently higher, and are consistent with the original lipidomics data gathered on C41 and Tuner systems used. As for the possibility of oxidized and artificially high reported CL levels, this was only a possibility and mentioned to show that the standards were checked prior resubmission for robustness. We believe that our values reported in the current manuscript version are accurate, and indeed more accurate than previously reported values

(depending on expression system and times used) for the reasons given in the original reply text. We hope this has cleared up any confusion.

3) Page 2, lines 74ff

The authors state 'For detergent (n-Decyl β -maltoside (DM), Glycon) purified samples, purification followed previously described protocols.

Please add a reference.

A reference to previous protocols has been added

In the original version the authors used DDM. Why was the detergent changed to DM?

There is no scientific reason for the switch. Our lab uses DM and DDM somewhat interchangeably for the solubilization of GlpG, and have found no difference in the activity or purity between samples. The reason for this change is to be more in line with previously published results, as DM-solubilised GlpG can be dialysed into liposomes for ssNMR studies. As the updated paper required a DM-solubilised ssNMR sample, it was thought best to stick to this detergent for all detergent solubilized samples in the work.

4) Line 142

Please add the pmol amounts of standard mixtures added to the extraction.

We have added the exact lipid standards and the relative pmol used in the materials and methods section.

Reviewer #3 (Remarks to the Author):

It has been well known that lipid exchange occurs between detergent-lipid micelles, liposomes, protein-encased nanodiscs, and SMALPs. The contribution of this study is the proposed concept that protein solubilized in DIBMALPs can also exchange among different protein-carrying DIBMALP populations via collisional mixing, which enables the study of protein-protein interactions (such as the catalytic cleavage of a protein substrate by GlpG shown in this study) under native environment of DIBMAPs (suppose the presumption that DIBMALPs represent native environment AND that the collisional mixing does not change such environment is true). The revised version added new data to mend some of the discrepancy shown in the original submission on GlpG expression and purification, activity assay, and lipidomic analysis. It does not provide new data to address the questions raised by Reviewer #3 regarding the structure of protein-carrying DIBMALPs and experimental evidence of protein migration via collisional mixing. Those are rather important questions that need to be answered with experimental data to support the authors' hypothesis. Let me explain why.

DIBMAs (or SMAs for that matter) are essentially a mixture of amphiphilic random copolymer chains of different detergency. The detergency of individual chains depends on its amphiphilic balance (i.e., DIB/MA ratio, sequence, and chain size). The earlier "cookie-cutter" model of how DIBMAs (or SMAs) solubilize membranes has been proven to be only a fantasy by recent data,

and for that reason, we should use DIBMALPs instead of nanodiscs or native nanodiscs or native-like nanodiscs unless we have evidence to show that we have successfully purified a portion of DIBMALPs (or SMALPs) to retrieve such a truly-existed population, because the solubilized membranes (i.e., DIBMALPs or SMALPs) is also a mixture of polymer-lipid self-assembled particles with different degrees of polymer-lipid inter-mixing corresponding to the different detergency of individual DIBMA chains as well as the polymer-to-lipid ratios used to solubilize the membrane. Although we all love the concept of detergent-free solubilization of membrane proteins, we need to be mindful that plenty of chains in any given DIBMA (or SMA) sample behave just like detergent because their individual detergency is strong.

We thank reviewer 3 for their insight, however disagree on a couple of points: The cookie-cutter theory that was initially proposed has long been known to be false, or as stated correctly by the reviewer a wish. Indeed, this submitted work never said anything akin to cookie cutter in-terms of particle stability. Additionally, whilst reviewer 3 is correct in stating that SMA (as well as some other polymers) are random co-polymers, the polymer used here is DIBMA – which is an alternating polymer of diisobutylene and maleic acid. The alternating nature means that, unlike SMA, there is no difference in detergency along the chain lengths. However, the manufacturer of DIBMA does produce a varied length mixture, which leads to different chain lengths and possibly a difference in detergency, but overall DIBMA's average chain length is relatively high compared to the usually used SMA (<https://smalp.net/polymers.html>).

The use of centrifugation to remove undissolved membranes and Ni-NTA resin to separate out the protein-carrying DIBMAPs as shown in this study doesn't necessarily produce nanodiscs; those selected protein-carrying DIBMAPs could well be a mixture of DIBMA-GlpG-lipid mixed micelles. In fact, previous study shown that the average GlpG-to-lipid ratio in DIBMAPs is ~1:10 (DOI: 10.1021/jacs.8b08441), which is very similar to what observed in detergent-solubilized GlpG (~1:14; DOI: 10.1016/j.jmb.2011.01.029) but far below what should be expected from nanodiscs with a diameter of 12-15 nm, suggesting a very large portion of protein-carrying DIBMALPs are actually in micelle forms.

We thank reviewer 3 for highlighting the previous literature – As a response to reviewer 2 we have attached in the SI the relative lipid:protein ratio per sample, which finds significantly more lipid in the DIBMALP samples than DM samples (SI Table 3). It is interesting however, that the previously published work has a low lipid:protein ratio (and indeed they question it in their text), and do not have a good explanation for their results. Having said that, the DLS performed by Barniol-Xicotá *et al.* show the expected hydrodynamic radius, of either discs or very large micelles, and the corresponding negative stain EM shows similar structures. We point reviewer 3 to the figure attached in this document (Figure 1) to show the shape and size of DIBMALPs produced by our hands, with the caveat that these are not the same preparations used for the manuscript.

Without separating and identifying the membrane-like DIBMALPs fractions from the micelle-like DIBMALPs fractions, the entire argument about protein hopping among different protein-carrying DIBMALP populations via collisional mixing stays as an unvalidated hypothesis because the fusion between micelles (DOI: 10.1021/jacs.7b09060) or between micelles and membranes (DOI: 10.1016/j.bpj.2021.04.012) could also explain the observed GlpG cleavage activity.

I'd like to propose the following suggestions for the authors to consider:

(1) Separate and identify the membrane-like DIBMALPs fractions.

Please show an SEC chromatogram of protein-carrying DIBMALPs. Show both dRI and UV-Vis signals. Calibrate the elution volume with protein standards of known hydrodynamic radius. At the very minimum, measure the GlpG-to-lipid ratio at a few selected elution volumes and explain which fractions will be collected for your study.

We thank the reviewer for their feedback, but believe that the suggested additional work by reviewer 3 is beyond the scope of this paper. As we believe that this amount of work is beyond the scope of the paper – where we show the use of mixing in polymer-based nanoparticles (commonly referred to as discs in the literature) to study the neighbouring environment of encapsulated membrane proteins. We have taken steps in the review process to tone down the use of the term discs, at reviewer request. We also submit the following figure (Figure 1) to satisfy the reviewer that the majority of the nanoparticles referred to in this paper are ‘discs’ (please see below), however we maintain that the disc nature or particle nature (as long as they are not micelle size) is the only important factor for the scope of this paper. We have also, at the request of reviewer 5, highlighted the issues with the literature name of nanodiscs in a footnote in the introduction.

The figure below shows a post Ni-NTA GlpG-DIBMA SEC profile (A), and calibration curve of known protein sizes (B) with the corresponding fractions to eluted GlpG-DIBMA displayed in SDS-PAGE gel (C). The large arrow denotes the fraction chosen for EM, which according to the column manual (S200 increase) has a hydrodynamic radius of 2.3nm (Ovalbumin). However, as can be seen in the corresponding negative stain EM (D), the particles observed are 1) broadly disc shaped, and 2) the expected size for DIBMA particles. This discrepancy is similarly noted for DLS data where hydrodynamic radius is calculated under the assumption of a globular protein (or such spherical system), whereas discs are not spherical. Whilst we note that the provided SEC and EM data are not a direct comparison, coming from a different expression of GlpG than that of the paper, the same methodology was used and we believe this should suffice to show that the protein containing particles are not purely micelles, but instead discs or perhaps larger aggregates (the latter fraction corresponds to excess imidazole from the Ni-NTA purification).

Additionally, (E) shows the negative EM image of another protein encapsulated in DIBMALPs, MthK. The authors note that this grid was prepared directly after Ni-NTA purification, in the same manner described in this work. As can be seen, the vast majority of the population is 1) (broadly) disc shaped, and 2) the expected size for MthK-DIBMALPs.

Figure 1 – SEC, SDS-PAGE, and negative stain EM of different DIBMA solubilisations. (A) SEC of GlpG-DIBMA post Ni-NTA purification of a different expression than that shown in the manuscript. Arrow corresponds to the fraction chosen for negative stain EM (D), with a corresponding hydrodynamic radius from standards of ovalbumin, 2.3nm. As calculated by calibration curve to known protein standards (B). (C) SDS-PAGE of fractions from (A), with expected protein bands for GlpG. (D) Negative stain EM of GlpG-DIBMA fraction A5.9 from (A,C), showing expected shape and size of DIBMA nanodiscs. (E) Negative stain EM of MthK-DIBMALPs, directly after Ni-NTA purification. Particles highlighted with white circle are believed to be a ‘top down’ view of MthK in DIBMALPs.

(2) The collisional mixing

At the bottom of any mixing (no matter how we call it) is the dissociation of species x from party A, diffuse of free x in the medium, and association of free x into party B. This could happen via the slow dissociation-association equilibrium of x between party A (or party B) and the free x in the medium, or the accelerated equilibrium of the same route, i.e., collisional mixing as originally coined by Nichols in the study of lipid transfer between detergent-lipid micelles (DOI: 10.1021/bi00411a006), where the collisional mixing was used to refer the accelerated rate of x dissociation from party A (or party B) - which is the rate-limiting step of the lipid mixing - due to collisions among party A and/or B.

We thank the reviewer again for their informative references. We however note that the Nichols paper referenced by the reviewer states in the conclusions that there are two forms of mixing that occur: “(1) *transient collisions or fusions which allow for rapid exchange of phospholipids and presumably other hydrophobic molecules and (2) a faster rate of phospholipid dissociation from the mixed micelles than from phospholipid vesicles*” (DOI: 10.1021/bi00411a006). The work proposed in this manuscript follows the current literature in proposing that the majority of the mixing occurs through collisional mixing (i.e. form 2, which was also described in a similar manner to Nichols *et al.* by Keller *et al.*). We are not ignoring the fact that lipids in the polymer nanoparticle systems could, in theory, diffuse from one nanoparticle or another, anymore than could occur in liposomes or detergent solutions. We are only proposing in this work that the *vast majority* of this collision for lipid transfer is through the collisional method. Similarly, the coincubation and cleavage of the integral membrane peptide TatA-FITC, or the SUMO-TatA-FLAG construct shows that the larger proteins are also transferred in solution, which is energetically unfavourable in a non-collision based hypothesis.

Keller *et al* (DOI: 10.1038/srep45875) showed that accelerated lipid transfer (i.e., collisional mixing) also happens among SMALPs when bulk solution concentrations of lipid in the donor and acceptor populations are high enough. As a side note for Keller’s work, the SMA with a S/MA=3/1 ratio was used, this variant of SMA is less popular in SMALP community because it is known to have strong detergency.

As such, a significant portion of SMALPs in that study is likely in polymer-lipid micelle forms, so the observed fast lipid transfer is in line with Nichols’ original study of lipid transfer between detergent-lipid micelles.

We agree with the reviewer that SMALPs with a 3:1 ratio do have a stronger solubilization efficiency than the more commonly used 2:1, whilst noting that Keller showed that this mixing also occurs for SMA 2:1 (10.1007/s00232-018-0024-0) and Glyco-DIBMA (10.1039/d1nr03811g).

The current study takes a leap of faith by proposing the collisional mixing of proteins among different protein-carrying DIBMALPs without any limitations (such as concentration or DIBMAP

structure) or supporting evidence besides the observed GlpG cleavage activity. Proteins don't magically hop between DIBMALPs. This proposal basically suggests that GlpG or its protein substrate can be released from one DIBMALP into buffer in their free forms via collision, then reinserting themselves into another DIBMALP.

We politely restate that we believe that from the vast work shown in the manuscript, and the literature precedent from Keller *et al* and others, that we believe collisional mixing of DIBMA (or other such polymer nanoparticles) is the major way that cargo such as lipids or membrane proteins are transferred from one particle to another (provided, as stated in the literature, that the concentration is high enough). We again restate that at no point do we confirm or hypothesize that the integral membrane proteins freely dissociate from one disc/particle to reinsert into another, but instead collisional mixing is the method of protein transfer.

Considering the large hydrophobic domain of both proteins and their strong interactions with the surrounding polymer-lipid matrix, this is a bold hypothesis and appears to be far less likely than a simple fusion of DIBMALPs. Without any experimental data to exclude the possibility of fusion among DIBMALPs, the observed cleavage activity is not sufficient to support the authors' hypothesis because fusions of different DIBMALPs could also bring GlpG close to its substrate.

We agree with the reviewer that DIBMALP fusion (and we believe re-separation) is the more likely method of cargo exchange over the free diffusion hypothesis, and believe that we have presented such a body of evidence to show for this work that this method of mixing can be utilized by other researchers for their studies. Hence the reworking of the text to rely more heavily on the methodology and application over the biophysical aspects of the mixing, as this has been previously described (quite thoroughly) by Keller *et al*.

(3) Unless I missed anything, I'm not sure how the authors concluded that GlpG and GlpGΔN extracted in DIBMALPs have 6x lipid:protein ratio than those extracted in DM (line 219-220). Please provide supporting data on that.

We apologise for this oversight, and have now added this data to the SI (SI Table 3) for clarification by any future readers.

I'm also not entirely sure what is the rationale to compare the activity between GlpG and GlpGΔN and what important discovery is drawn out of this comparison. I recommend the authors to explain the rationale of this experimental design in the abstract, and to elaborate the important discovery, if any, in the summary.

The comparison between GlpG and GlpGΔN is a remnant of the original aim of the work, which was to see if GlpG and the N-terminal domain deficient GlpG interact differently with the membrane or neighbouring lipid. As has been shown previously, GlpG is able to diffuse rapidly through the membrane, and has a different orientation in the membrane depending on the N-terminal domain removal (DOI: doi:10.1126/science.aaa0076). It has also been shown through HDX-MS (DOI:10.1002/anie.20170965), and some preliminary studies of ours through ssNMR) that the N-terminal domain interacts directly with the membrane. The original goal of this work was to see if the 'natively extracted' DIBMA could highlight specific differences between GlpG and GlpGΔN, and if the mixing could be used to identify specifically retained lipids. Due to wording limitations we cannot amend the text of the abstract to include this, but have added a few sentences to include this in the introduction.

(4) There are a number of habitual talking points in the DIBMALP or SMALP community that are not entirely true, I'm listing a few examples below for the authors to reconsider:

- Title: ... to probe the native environment of...

Unless I missed anything, I don't see any evidence about the native environment enabled by DIBMALPs. The authors' own lipidomic analysis confirmed that no significant lipid composition difference is observed between the DM- and DIBMA-solubilized GlpG (Fig 2), and no specific lipid species is observed to be tightly bound to DM- and DIBMA-solubilized GlpG, or retained after DMPC or POPC washing (Fig. 3).

The title was decided due to the literature precedent to involve 'native' or 'native-like' to differentiate between different nanodiscs or nanoparticle types. Some examples of this in the literature are: [10.1002/anie.201610441](https://doi.org/10.1002/anie.201610441), [10.3390/nano11071771](https://doi.org/10.3390/nano11071771), [10.1038/s41598-018-33208-1](https://doi.org/10.1038/s41598-018-33208-1), [10.1039/d1nr03811g](https://doi.org/10.1039/d1nr03811g), [10.1038/s41565-023-01538-5](https://doi.org/10.1038/s41565-023-01538-5), [10.1016/j.bpj.2023.11.021](https://doi.org/10.1016/j.bpj.2023.11.021), [10.1038/s41467-023-39763-0](https://doi.org/10.1038/s41467-023-39763-0), [10.1016/j.bbamem.2023.184143](https://doi.org/10.1016/j.bbamem.2023.184143), [10.1021/acs.biomac.2c00935](https://doi.org/10.1021/acs.biomac.2c00935).

- Line 30-31: ... the use of detergent in protein extraction and purification leads to a loss of the native lipid environment...

Extract of proteins from native membranes by any means, SMA or DIBMA included, would lead to a loss of the native lipid environment of the proteins. Period.

Again, we politely disagree with the comparison between detergent extraction and polymer-based extraction being similarly disruptive. Whilst true that any extraction leads to a loss of native environment, there is a clear difference shown in the literature that detergent extraction vs polymer extraction are not the same in terms of disruption, particularly when showing membrane protein stability (DOI: [10.1002/anie.201610778](https://doi.org/10.1002/anie.201610778)) and activity (DOI: [10.1021/nl3020395](https://doi.org/10.1021/nl3020395)).

- Line 36-37: ...Lipid-polymer nanodiscs, such as SMALPs and DIBMALPs, have been shown to extract integral membrane proteins along with their native lipids into nanodiscs, without the use of detergents at any step ...

Again, SMALPs and DIBMALPs are "polymer-lipid nanoparticles" (original definition) instead of "nanodiscs" because of their diverse structures; SMA and DIBMA themselves could have detergency similar to detergents and behave just like detergents; detergents are also known to extract integral membrane proteins along with their native lipids as shown in many protein crystallography studies as well as the authors' own data (Fig 2 & 3).

We have amended this text, as it was a remnant from the original draft. We have also added a footnote explaining the difficulties in the literature surrounding nanodiscs vs nanoparticles, which was suggested by reviewer 5.

- Line 38-39: ...Proteins within these discs often possess increased stability compared to their detergent solubilized counterparts...

I don't see any systematic study to validate this generalized statement. Just like SMA or DIBMA that can be made with different detergency, many different detergents exist with different detergency too. We may compare the stability of a specific polymer-solubilized proteins with a

specific detergent-solubilized proteins, but a generalized statement such as this one does not make any sense.

Whilst we agree that no large scale systematic study has been performed (to our knowledge), there have been some attempts of studying stability of proteins in these systems compared to detergent (10.1016/j.eurpolymj.2018.12.008), as well as several individual comparisons made (10.1021/acs.nanolett.0c04911, references 10-15, and 10.1039/d1nr03811g, references 6, 10, 13, 15-22).

Reviewer #5 (Remarks to the Author):

Review of “Using collisional mixing in lipid-polymer nanoparticles to probe the native environment of intra-membrane rhomboid protease GlpG and its associated activity” by H. Sawczyk et al.

Given that this paper has already undergone extensive review, I will limit my commentary to the discussion between the authors and reviewer 3.

This paper presents evidence of lipid mixing in a DIBMALP containing protein. I believe that the community has been aware of such lipid mixing for a while, in fact, it was used by Edler et al. (see <https://doi.org/10.1016/j.bbadv.2021.100033>) to exchange deuterated lipids into an OMPF containing SMALP in at least 2021. However there has been no systematic study of this process and as such this paper certainly merits publication on that basis alone. The paper also suggests that proteins can mix within different populations of DIBMALPs which allows for studies of protein-protein interactions as illustrated here. This is also a very interesting result and adds to the case for publication.

That said, reviewer 3 is making an extremely valid point that the simplistic view of these particles as nanodiscs is not really accurate (this is nicely demonstrated in the article <https://doi.org/10.1021/acs.biomac.3c00034> that was referenced by reviewer 3 previously) and that the authors are lacking some evidence to support their mechanistic conclusions, particularly on collisional mixing to exchange proteins. I agree with reviewer 3 that the authors have not provided sufficient evidence that this is the mechanism of exchange, although it is not an unreasonable hypothesis based on the literature.

We thank reviewer 5 for coming in and reviewing at such a short notice (as well as their positive review overall), and thank them for their reference which was missed in the initial literature search – the introduction has been amended to reflect this paper.

Reviewer 3 is suggesting that the authors should obtain substantially more experimental evidence to support their claims. I think this is unnecessary since that will clearly add a substantial delay to publication. In my view the result that lipids and proteins can exchange between DIBMALPs (whatever their form) is already sufficiently interesting for publication and that the detail of structure and mechanism would be suitable for future publications. My suggestion to resolve the impasse between the authors and reviewer 3 would be for the authors to tone down their description and add appropriate caveats. For example, use of the term

nanodisc could be qualified in a referenced footnote that explains the known variability in these particles and the cartoons shown in figs 1 & 5 could add a similar caveat.

This caveat has been added where requested, and we hope that it is sufficiently worded for all parties.

Similarly, to avoid propagating the other misconceptions within the SMALP/DIBMALP community that are highlighted by the reviewer, I suggest a careful rephrasing of the introduction. I do think that the authors should be free to make an hypothesis regarding the exchange mechanism, as long as it is clear that this is their opinion and that the evidence for structure and mechanism within their study is limited. The addition of a short discussion and a reference to other possible mechanisms such as micelle fusion as suggested by reviewer 3 would certainly not detract from the main conclusions of the paper.

We once again thank reviewer 5 for their input, and have modified the introduction to refer to 'collisional lipid mixing' as proposed by Keller *et al.* to avoid confusion that this paper is definitive proof of the mechanism, and instead focuses more on the use of the phenomenon for the study of membrane protein activity in these systems.

REVIEWER COMMENTS

Reviewer #1 (Remarks to the Author):

I have examined the updates made to the manuscript, focussing on changes and comments made in response to my previous review. Most of the responses are appropriate and satisfactory, with the inclusion of a complete data table showing results of lipid and protein quantitation being particularly helpful. However, there are some questions that remain and need to be addressed and/or fixed prior to publication, as outlined below.

In the previous version of the manuscript Figure 4A showed activity data from biological replicates that was of questionable quality because the sample appears to have suffered from significant degradation (as shown in Fig. S1 C). The authors removed this data from the main text of the paper according to reviewer suggestion, but the data was retained in the Supplementary material. The figure legend for Figure S4 should indicate that only the data from panels A and C are shown for the activity data in Figure 4.

It is a little misleading to state that the gel-based activity results are 'in-line with previously published results'. The quality of the previous results that is being referred to is significantly higher and more convincing than the results shown in the Supplementary Material. More concerning is the data shown for DIBMA-solubilized ΔN , as the activity is not consistent with previous results since there is evidence of non-specific cleavage in the control using the S201A knockout mutant and either very weak or no intensity change for the GlpG-associated cleavage product for many of the samples. The authors should remove the in-gel activity data for ΔN if they do not have more convincing data to show. In that case it would be more appropriate to state that gel-based results for FL GlpG in DIBMA show some evidence for cleavage that was not seen for the knockout mutant.

As a side note, the authors should make sure that the labeling of biological replicates is consistent across all figures over the entire manuscript. Specifically, TAMRA-labeling of FL1 and FL2 in Fig. S1D seems to match FL 2 and FL1 in the purified samples shown in Fig. S1C. Same issue for ΔN lanes in this figure.

In the rebuttal the authors explain that both bands encompassed by the rectangle labeled 2* in Figure S1C were analyzed by MS, and only the identity of the most abundant species was reported. This is confusing as they state this is the 50S ribosomal protein L13, even though one of these bands should be GlpG ΔN . If neither of these bands was identified as GlpG ΔN , then this gives the impression that the sample did not contain any of the purified target protein. I assume that this was not the case, so clarification is needed. (If no GlpG ΔN was detected, why not?)

Reviewer #2 (Remarks to the Author):

The authors have gone to great lengths to carefully address the many points of criticism and suggestions made by the reviewers. In view of the fact that the review process and thus the different viewpoints especially of reviewer 3 and the authors are made available to the community, I would like to propose that the manuscript be accepted. Personally, I think the exchange between the reviewers and the authors is very helpful and instructive and shows that there are very different views on the topic. Especially because of this fact, I consider a methodological paper that addresses these questions to be very interesting. I think the major discrepancies between reviewer 3 and the authors should be picked up in the discussion and will certainly be valuable information for the community.

Reviewer #3 (Remarks to the Author):

I'd like to first applaud the authors' persistence in the past year to continuously supply new data in response to the reviewers' request, which significantly improved the quality of this paper. The original submission of this paper was set to show the importance of "the native environment" (a commonly fantasized concept in SMALPs/DIBMALPs community) on the activity of GlpG. Following the authors' own lipidomic analysis and lipid exchange assays, it turned out there is no such "native environment" in DIBMA-extracted GlpG that purportedly plays the critical role on its activity. The authors have made the much needed corrections/clarifications in the latest submission except for a few important points below, which have to be addressed to win my vote.

(1) Title. As I pointed out in my last review, the use of "the native environment" in the title is a habitual talking point but is unfortunately not supported by the authors' own data. For example, Fig. 2 clearly shows no significantly different lipid composition exists between crude membranes, DM- and DIBMA-solubilized GlpG, and Fig.3 clearly shows no specific GlpG-lipid interaction exists in either DM- or DIBMA-solubilized GlpG. In addition, the GlpG activity is not reduced when exogenous lipids were "washed in". It does not matter how many other papers used "the native environment" as a talking point; using this catching phrase while knowingly understand it is not supported by the authors' own data is misleading.

(2) Collision mixing. The most important contribution of this work, according to the authors, is the discovery that "the collisional lipid mixing" reported earlier by Keller et al should be renamed "collisional mixing" because the transfer of lipids among DIBMALPs (due to collision) also applies to the proteins carried by DIBMALPs. This argument suggests that proteins can somehow magically hop between DIBMALPs due to collision, which is not true as I pointed out in my last review. It is

easy to get lost in translation by playing the word games without looking into the underlying physics behind this concept. The original concept of collision mixing reported by Nichols (DOI: 10.1021/bi00411a006) proposed two underlying mechanisms as the authors correctly cited in their rebuttal letter. For the first mechanism (i.e., “transient collisions or fusions”), Nichols further elaborated that “Presumably, fusion of two micelles destabilizes the balance of energetic forces and geometric constraints that determines the micellar size, and the fused micelle rapidly divides to return to a more stable size... the exchange of phospholipids during the transient fusion complexes does not require exposure of the hydrophobic acyl chains of the phospholipids to the water. Formation of these transient fusion complexes may provide one component of the “hydrocarbon continuum” that has been postulated to explain the intestinal absorption of very hydrophobic lipid molecules.” In Keller’s paper (DOI: 10.1038/srep45875), it also proposed that the collisional lipid transfer among SMALPs “imply that lipids exchange through a “hydrocarbon continuum” enabled by the flexible nature of the SMA belt surrounding the lipid-bilayer core.” It’s clear from prior studies that the fast exchange of SMALPs or DIBMALPs components can be attributed to the transient fusion (and subsequent separation) of two micellar complexes that occurs through a “hydrocarbon continuum” regardless of whether this component is a lipid or a protein. What the authors observed in this work is not beyond what Keller reported earlier, and to a larger extent what Nichols proposed in 1980s, as they all describe the same physical phenomenon. Instead of introducing a new term of “collisional mixing” to suggest proteins can hop among DIBMALPs due to collision, which is not true and misleading to those who are not familiar with literatures, I suggest the authors to cite Nichols earlier work, tone down the so-called difference between this work and Keller’s paper, and elaborate the underlying physics behind this concept.

(3) The issue of “nanodisc”. Thanks for the additional data provided as Fig 1 in the authors’ rebuttal letter, it should become clear to everyone that the GlpG-carrying DIBMALPs (even after Ni-NTA purification) have a wide distribution of structures with different sizes, and the term “nanodiscs” is a gross misinterpretation of DIBMALPs. The negative-staining TEM (Fig. 1D and E, rebuttal letter) by no means supported the nanodisc claim by the way – the 2D projected nanoparticle “blobs” in EM can be anything: polymer micellar aggregates, polymer-lipid mixed micelles, etc. (again, there are papers that took these EM “blobs” as evidence of “nanodiscs”, but anyone with EM expertise would not endorse that). The authors did take advice from Review #5 to add a footnote that SMALPs and DIBMALPs are “commonly referred to as ‘nanodiscs’”. This footnote is necessary but still confuses audience at times. For example, line 36-37, “Lipid-polymer nanoparticles have been shown to extract integral membrane proteins into nanodiscs”?? My recommendation is simple: unless the authors collected data from a carefully purified DIBMALPs fraction that has sufficient evidence to support it is indeed nanodisc (which is not the case here), I’d suggest the authors to replace all the “nanodiscs” with “polymer-lipid nanoparticles” or “DIBMALPs” when possible. Calling DIBMALPs “nanodiscs” without supporting evidence is misleading. It does not matter whether other papers have misused this term or not.

(4) GlpG-lipid ratio in DIBMALPs. I applaud the authors effort to show new data that compare the GlpG-lipid ratios in DM micelles and DIBMALPs (Table S3). There is only one caveat though: the

authors need specify whether this measurement was done using one specific SEC fraction of DIBMALPs, or all DIBMALPs collected after Ni-NTA purification. The comparison of GlpG-lipid ratios will be meaningless and misleading without specifying the samples to start with because many different DIBMALPs structures exist after Ni-NTA purification (Fig. 1, rebuttal letter). For example, fractions 5A.1-5A.4 that approach the exclusion limit of the SEC (i.e., Thyroglobulin) identified as incompletely dissolved membrane components (i.e., polymer-remodeled liposomes or proteoliposomes, DOI: 10.1021/acs.biomac.3c00034) will artificially increase the lipid-to-GlpG ratio if no fractionation effort was made. If all DIBMALPs collected after Ni-NTA purification was used to determine the GlpG-lipid ratio, the authors need add a discussion on the limitation of what this ratio implies when it is compared with that of DM micelles.

(5) Despite my earlier critiques in the last review, the authors insist to claim that “proteins within SMALPs or DIBMALPs often possess increased stability compared to their detergent solubilized counterparts” (line 38-40). As I explained earlier, this blanket claim without specifying a particular protein, a particular detergent micelle system, or a particular polymer-solubilized state is meaningless, highly misleading to those who are not familiar with literatures, and unsubstantiated in the first place. Allow me to dive a little bit deeper into this widely cross-cited but largely half-truth claim.

The authors cited Ref #6 and 7 as support. Ref #6 did not provide any evidence about the “increased stability” of proteins; Ref #7 did not either but cited three additional papers to make the same claim: The 1st paper (DOI: 10.1021/bi1003712) reported the nonannular lipid binding sites on KcsA increase its stability. This paper has nothing to do with SMALPs or DIBMALPs by the way; the 2nd paper (DOI: 10.1021/nl3020395) used EPR to show that in Lipodisq particles “the structural and dynamic integrity of bR was retained when compared with data for bR obtained in the native membrane and in detergents and then with crystal data”. No evidence was provided to show the “increased stability” of bR in Lipodisq as compared to detergent micelles; the 3rd paper (DOI: 10.1016/j.bbamem.2019.183152) reported that Dopamine receptors (DRs) extracted in Lipodisq particles maintain their activity. It should be pointed out that active Dopamine receptors (DRs) have been extracted routinely in detergent micelles with the right choice of detergents. No evidence was provided to show the “increased stability” of Dopamine receptors (DRs) in Lipodisq as compared to that in detergent micelles. Taken together, Ref #6 and 7 can’t support the blanket claim about the “increased stability” of proteins in DIBMALPs or SMALPs.

Another place the authors made a similarly misleading claim is in line 54-56: “GlpG has previously been solubilized by both SMA and DIBMA, showing more native like activity with fluorophosphonate probe labelling (TAMRA-FP) and enhanced stability than detergent solubilized GlpG17,18.” Although the authors cited both Ref #17 and 18 as support, only Ref #18 is relevant to the particular claim made here. In Ref #18 it was shown that the DDM-solubilized GlpG is 8x more active (as judged by FP-Rh labeling) than GlpG in native membrane pellets or extracted in DIBMALPs (!), and the latter two have similar level of low GlpG activity. It was speculated in Ref #18 that this difference could be

due to the “increased mobility or active site accessibility” of GlpG in DDM micelles. In another word, FP-Rh can’t gain access to its target site on GlpG in native membrane pellets or DIBMALPs as easily as that in DDM micelles. The authors interpreted this potential limitation of the activity assay as evidence of GlpG in DIBMALPs “showing more native like activity” (just because it has a similarly low level of activity as native membrane pellets?) is highly misleading and has to be revised. Ref #18 also reported that VcROM, a rhomboid protease of *Vibrio cholerae*, did not undergo autoprocessing (i.e., targeting itself as a substrate) in DIBMALPs as it does in DDM micelles. This observation can’t be used to support the claim that GlpG has “enhanced stability” in DIBMALPs because the activity and stability is not necessarily correlated (note the GlpG in DDM micelles is 8x more active than in DIBMALPs as reported in the same reference).

Reviewer #5 (Remarks to the Author):

In my opinion the authors have made appropriate changes based on my previous review and the article is now acceptable for publication.

We once again would like to thank the reviewers for their patience and helpful comments throughout this extended review process. Below are specific responses to the points raised by the reviewers, where relevant. Amendments to the manuscript as requested by the reviewers are highlighted in the text in yellow. There is an additional amendment in green due to the authors rewording of a confusing sentence.

REVIEWER COMMENTS

Reviewer #1 (Remarks to the Author):

I have examined the updates made to the manuscript, focussing on changes and comments made in response to my previous review. Most of the responses are appropriate and satisfactory, with the inclusion of a complete data table showing results of lipid and protein quantitation being particularly helpful. However, there are some questions that remain and need to be addressed and/or fixed prior to publication, as outlined below.

In the previous version of the manuscript Figure 4A showed activity data from biological replicates that was of questionable quality because the sample appears to have suffered from significant degradation (as shown in Fig. S1 C). The authors removed this data from the main text of the paper according to reviewer suggestion, but the data was retained in the Supplementary material. The figure legend for Figure S4 should indicate that only the data from panels A and C are shown for the activity data in Figure 4.

We thank Reviewer 1 for their keen eye, and have subsequently amended SI Fig. 4 to state that only A and C were analysed for the rates shown in Fig. 4.

It is a little misleading to state that the gel-based activity results are 'in-line with previously published results'. The quality of the previous results that is being referred to is significantly higher and more convincing than the results shown in the Supplementary Material. More concerning is the data shown for DIBMA-solubilized ΔN , as the activity is not consistent with previous results since there is evidence of non-specific cleavage in the control using the S201A knockout mutant and either very weak or no intensity change for the GlpG-associated cleavage product for many of the samples. The authors should remove the in-gel activity data for ΔN if they do not have more convincing data to show. In that case it would be more appropriate to state that gel-based results for FL GlpG in DIBMA show some evidence for cleavage that was not seen for the knockout mutant.

We have amended the text in the main manuscript to state that the results from the fluorescent labelling is 'broadly in line with previously published results' (which is true), and have pointed out that the gel-based assay of SUMO-TatA-FLAG is of non-ideal quality. We however would like to keep the gel assay for both ΔN and FL GlpG in the SI for completeness, and to not raise questions from readers about the location of ΔN in this assay suite.

As a side note, the authors should make sure that the labeling of biological replicates is consistent across all figures over the entire manuscript. Specifically, TAMRA-labeling of FL1 and FL2 in Fig. S1D seems to match FL 2 and FL1 in the purified samples shown in Fig. S1C. Same issue for ΔN lanes in this figure.

We have since checked the FL1 and FL2 labelling and have subsequently changed the labelling for SI Fig.1C to match the pattern of bands observed in D. However, the ΔN sample labelling appears to be correct in the authors opinion.

In the rebuttal the authors explain that both bands encompassed by the rectangle labeled 2* in Figure S1C were analyzed by MS, and only the identity of the most abundant species was reported. This is confusing as they state this is the 50S ribosomal protein L13, even though one of these bands should be GlpG ΔN . If neither of these bands was identified as GlpG ΔN , then this gives the impression that the

sample did not contain any of the purified target protein. I assume that this was not the case, so clarification is needed. (If no GlpG Δ N was detected, why not?)

We apologise for the confusion in our response, but the trypsin digest does show a high concentration of both GlpG and GlpG Δ N in the expected bands, which is stated in the SI Fig. 1 caption. We believe this confusion is due to the numbering of the bands sent to MS, as well as the numbering of the top 5 highest-concentration of proteins received from MS. For clarity of future readers, we have highlighted the relevant GlpG in this caption in bold.

Reviewer #2 (Remarks to the Author):

The authors have gone to great lengths to carefully address the many points of criticism and suggestions made by the reviewers. In view of the fact that the review process and thus the different viewpoints especially of reviewer 3 and the authors are made available to the community, I would like to propose that the manuscript be accepted. Personally, I think the exchange between the reviewers and the authors is very helpful and instructive and shows that there are very different views on the topic. Especially because of this fact, I consider a methodological paper that addresses these questions to be very interesting. I think the major discrepancies between reviewer 3 and the authors should be picked up in the discussion and will certainly be valuable information for the community.

We thank reviewer 2 again for their helpful (and positive) comments, and will take this recommendation for answering reviewer 3's comments.

Reviewer #3 (Remarks to the Author):

I'd like to first applaud the authors' persistence in the past year to continuously supply new data in response to the reviewers' request, which significantly improved the quality of this paper. The original submission of this paper was set to show the importance of "the native environment" (a commonly fantasized concept in SMALPs/DIBMALPs community) on the activity of GlpG. Following the authors' own lipidomic analysis and lipid exchange assays, it turned out there is no such "native environment" in DIBMA-extracted GlpG that purportedly plays the critical role on its activity. The authors have made the much needed corrections/clarifications in the latest submission except for a few important points below, which have to be addressed to win my vote.

(1) Title. As I pointed out in my last review, the use of "the native environment" in the title is a habitual talking point but is unfortunately not supported by the authors' own data. For example, Fig. 2 clearly shows no significantly different lipid composition exists between crude membranes, DM- and DIBMA-solubilized GlpG, and Fig.3 clearly shows no specific GlpG-lipid interaction exists in either DM- or DIBMA-solubilized GlpG. In addition, the GlpG activity is not reduced when exogenous lipids were "washed in". It does not matter how many other papers used "the native environment" as a talking point; using this catching phrase while knowingly understand it is not supported by the authors' own data is misleading.

We once again disagree with the reviewer over the necessity of changing the title, in light of the literature precedent of this term, and that we have added caveats to the composition in our text since the initial submission. We would also like to restate that whilst there are no significant compositional changes between DM- and DIBMA-solubilised protease, there is a difference in lipid:protein ratio which is important for the discussion of indirect effects on protein function. Additionally, we would like to restate that a change in protease activity (not necessarily a decrease or increase, but a change) was observed upon exogenous lipid addition. This finding is in line with the previous findings of Engberg *et al.* for refolded GlpG compared to detergent-solubilised GlpG, and neither work can yet state why this change is (although some theories come to mind, but are withheld to not over-extend the manuscript in scope away from the methodology shown).

(2) Collision mixing. The most important contribution of this work, according to the authors, is the discovery that “the collisional lipid mixing” reported earlier by Keller et al should be renamed “collisional mixing” because the transfer of lipids among DIBMALPs (due to collision) also applies to the proteins carried by DIBMALPs. This argument suggests that proteins can somehow magically hop between DIBMALPs due to collision, which is not true as I pointed out in my last review. It is easy to get lost in translation by playing the word games without looking into the underlying physics behind this concept. The original concept of collision mixing reported by Nichols (DOI: 10.1021/bi00411a006) proposed two underlying mechanisms as the authors correctly cited in their rebuttal letter. For the first mechanism (i.e., “transient collisions or fusions”), Nichols further elaborated that “Presumably, fusion of two micelles destabilizes the balance of energetic forces and geometric constraints that determines the micellar size, and the fused micelle rapidly divides to return to a more stable size... the exchange of phospholipids during the transient fusion complexes does not require exposure of the hydrophobic acyl chains of the phospholipids to the water. Formation of these transient fusion complexes may provide one component of the “hydrocarbon continuum” that has been postulated to explain the intestinal absorption of very hydrophobic lipid molecules.” In Keller’s paper (DOI: 10.1038/srep45875), it also proposed that the collisional lipid transfer among SMALPs “imply that lipids exchange through a “hydrocarbon continuum” enabled by the flexible nature of the SMA belt surrounding the lipid-bilayer core.” It’s clear from prior studies that the fast exchange of SMALPs or DIBMALPs components can be attributed to the transient fusion (and subsequent separation) of two micellar complexes that occurs through a “hydrocarbon continuum” regardless of whether this component is a lipid or a protein. What the authors observed in this work is not beyond what Keller reported earlier, and to a larger extent what Nichols proposed in 1980s, as they all describe the same physical phenomenon. Instead of introducing a new term of “collisional mixing” to suggest proteins can hop among DIBMALPs due to collision, which is not true and misleading to those who are not familiar with literatures, I suggest the authors to cite Nichols earlier work, tone down the so-called difference between this work and Keller’s paper, and elaborate the underlying physics behind this concept.

We thank the reviewer once again for highlighting this paper, which is now reference 35 in the main text (discussion).

We have since amended the discussion to include the terminology as follows: “This so-called ‘hydrocarbon continuum’ has been shown to occur in both detergent micelles and lipid-polymer nanoparticles systems^{20,36}, and facilitate a fast transfer of lipids across this continuum.” And “Additionally, we show for the first time through the cleavage of integral membrane protein TatA by GlpG that not only does this mixing through the ‘hydrophobic continuum’ occur not only for lipids, but for two protein-containing DIBMALPs.” We believe this change is sufficient, as this methodology work is novel for the protein involvement in the mixing (hence the terminology change proposed). We do not agree that we need to cover the underlying physics of this application, as it is so thoroughly covered by both Nichols and Keller.

(3) The issue of “nanodisc”. Thanks for the additional data provided as Fig 1 in the authors’ rebuttal letter, it should become clear to everyone that the GlpG-carrying DIBMALPs (even after Ni-NTA purification) have a wide distribution of structures with different sizes, and the term “nanodiscs” is a gross misinterpretation of DIBMALPs. The negative-staining TEM (Fig. 1D and E, rebuttal letter) by no means supported the nanodisc claim by the way – the 2D projected nanoparticle “blobs” in EM can be anything: polymer micellar aggregates, polymer-lipid mixed micelles, etc. (again, there are papers that took these EM “blobs” as evidence of “nanodiscs”, but anyone with EM expertise would not endorse that). The authors did take advice from Review #5 to add a footnote that SMALPs and DIBMALPs are “commonly referred to as ‘nanodiscs’”. This footnote is necessary but still confuses audience at times. For example, line 36-37, “Lipid-polymer nanoparticles have been shown to extract integral membrane proteins into nanodiscs”?? My recommendation is simple: unless the authors collected data from a carefully purified DIBMALPs fraction that has sufficient evidence to support it is indeed nanodisc (which is not the case

here), I'd suggest the authors to replace all the "nanodiscs" with "polymer-lipid nanoparticles" or "DIBMALPs" when possible. Calling DIBMALPs "nanodiscs" without supporting evidence is misleading. It does not matter whether other papers have misused this term or not.

We, once again, politely disagree with the reviewer that the broad term of nanodiscs to refer to those purified (through removal of excess polymer and polymer-lipid micelles) lipid-polymer nanoparticles (protein encapsulating or not) is inaccurate. We also disagree that a significantly mixed population is present in the EM data shown previously (review reply). The aggregation observed and noted by the reviewer is a common charge and stain-dependent aggregation, and is an artefact of negative stain EM in particular. We believe that the distinction within the footnote to describe the contentious nature of the broad nanodiscs term, as well as the flexible use of particle, DIBMALP, and nanodiscs where grammar and sentence flow dictates is sufficiently clear to future readers. As pointed out by other reviewers, the discussion in this review process being available publicly will also add to any reader's context on this terminology in the paper and the wider field.

(4) GlpG-lipid ratio in DIBMALPs. I applaud the authors effort to show new data that compare the GlpG-lipid ratios in DM micelles and DIBMALPs (Table S3). There is only one caveat though: the authors need specify whether this measurement was done using one specific SEC fraction of DIBMALPs, or all DIBMALPs collected after Ni-NTA purification. The comparison of GlpG-lipid ratios will be meaningless and misleading without specifying the samples to start with because many different DIBMALPs structures exist after Ni-NTA purification (Fig. 1, rebuttal letter). For example, fractions 5A.1-5A.4 that approach the exclusion limit of the SEC (i.e., Thyroglobulin) identified as incompletely dissolved membrane components (i.e., polymer-remodeled liposomes or proteoliposomes, DOI: 10.1021/acs.biomac.3c00034) will artificially increase the lipid-to-GlpG ratio if no fractionation effort was made. If all DIBMALPs collected after Ni-NTA purification was used to determine the GlpG-lipid ratio, the authors need add a discussion on the limitation of what this ratio implies when it is compared with that of DM micelles.

We thank the reviewer for the feedback, which involves the data from the Ni-NTA purified DIBMALPs, as stated in the main text. We believe that this data is not meaningless however, as the lipid (and protein) population sampled is compositionally in a hydrophobic continuum, as stated by Nichols and the reviewer themselves. Removal or further purification by SEC to obtain a 'true' sample of GlpG-DIBMA nanodiscs and the corresponding lipid:protein ratio is functionally meaningless in this case, as would be inaccurate to the subsequent comparison with DM micelles that have not been purified in the same manner. We believe the comparison currently in the manuscript is a fair comparison between DM and DIBMA solubilized protease in the same methodology.

(5) Despite my earlier critiques in the last review, the authors insist to claim that "proteins within SMALPs or DIBMALPs often possess increased stability compared to their detergent solubilized counterparts" (line 38-40). As I explained earlier, this blanket claim without specifying a particular protein, a particular detergent micelle system, or a particular polymer-solubilized state is meaningless, highly misleading to those who are not familiar with literatures, and unsubstantiated in the first place. Allow me to dive a little bit deeper into this widely cross-cited but largely half-truth claim.

The authors cited Ref #6 and 7 as support. Ref #6 did not provide any evidence about the "increased stability" of proteins; Ref #7 did not either but cited three additional papers to make the same claim: The 1st paper (DOI: 10.1021/bi1003712) reported the nonannular lipid binding sites on KcsA increase its stability. This paper has nothing to do with SMALPs or DIBMALPs by the way; the 2nd paper (DOI: 10.1021/nl3020395) used EPR to show that in Lipodisq particles "the structural and dynamic integrity of bR was retained when compared with data for bR obtained in the native membrane and in detergents and then with crystal data". No evidence was provided to show the "increased stability" of bR in Lipodisq as compared to detergent micelles; the 3rd paper (DOI: 10.1016/j.bbamem.2019.183152) reported that Dopamine receptors (DRs) extracted in Lipodisq particles maintain their activity. It should be pointed out that active Dopamine receptors (DRs) have been extracted routinely in detergent micelles with the right

choice of detergents. No evidence was provided to show the “increased stability” of Dopamine receptors (DRs) in Lipodisq as compared to that in detergent micelles. Taken together, Ref #6 and 7 can’t support the blanket claim about the “increased stability” of proteins in DIBMALPs or SMALPs.

We politely disagree, again, with reviewer 3 about the broad statement that lipid-polymer encapsulation increases membrane protein stability compared to detergent counterparts is wholly incorrect or inaccurate. We believe that the cited references given in the previously reply are sufficient for such a general statement (in the context of this work) that detergent micelles are less stable environments for membrane proteins than lipid-polymer nanoparticles, when it comes to thermo-stability and native-like function. We agree that the first cited paper from reference 7 does not directly relate to this stability, but the papers directly cited by the authors of this manuscript in the previous reply do.

More specifically, the structural and dynamic integrity of bR and stability is shown by Orwick *et al*, the absorption shift noted in Figure 2 is indicative of a native-like function, and has previously been shown to correspond to the hydrophobicity of the retinal pocket (itself a function of the proteins dynamics and fold). A shift such as that observed for β -OG detergent is indicative of a dynamic protein. A similar conclusion can be drawn from the DR solubilization, where stated in said paper that dopamine receptors often have to have thermostabilised mutations to enable detergent purification and functional characterization or structural determination.

Again, the authors agree that each protein, detergent, and polymer system must be considered for any specific case and further work is needed for this (outside of the scope of this specific manuscript). However, there is a broad trend in the literature that shows that polymer solubilisation leads to more native-like function, structure, and dynamics, and therefore stability compared to detergent counterparts. We again highlight that this comment or disagreement is broadly out of scope of the work of this manuscript, and is only used to highlight the more native-like environment (broadly speaking) than detergent micelles for membrane protein research.

Another place the authors made a similarly misleading claim is in line 54-56: “GlpG has previously been solubilized by both SMA and DIBMA, showing more native like activity with fluorophosphonate probe labelling (TAMRA-FP) and enhanced stability than detergent solubilized GlpG17,18.” Although the authors cited both Ref #17 and 18 as support, only Ref #18 is relevant to the particular claim made here. In Ref #18 it was shown that the DDM-solubilized GlpG is 8x more active (as judged by FP-Rh labeling) than GlpG in native membrane pellets or extracted in DIBMALPs (!), and the latter two have similar level of low GlpG activity. It was speculated in Ref #18 that this difference could be due to the “increased mobility or active site accessibility” of GlpG in DDM micelles. In another word, FP-Rh can’t gain access to its target site on GlpG in native membrane pellets or DIBMALPs as easily as that in DDM micelles. The authors interpreted this potential limitation of the activity assay as evidence of GlpG in DIBMALPs “showing more native like activity” (just because it has a similarly low level of activity as native membrane pellets?) is highly misleading and has to be revised. Ref #18 also reported that VcROM, a rhomboid protease of *Vibrio cholerae*, did not undergo autoprocessing (i.e., targeting itself as a substrate) in DIBMALPs as it does in DDM micelles. This observation can’t be used to support the claim that GlpG has “enhanced stability” in DIBMALPs because the activity and stability is not necessarily correlated (note the GlpG in DDM micelles is 8x more active than in DIBMALPs as reported in the same reference).

The authors would like to make some GlpG or rhomboid protease-specific clarifications: The labelling by FP-TAMRA is indeed indicative of active site accessibility to the solvent (aqueous buffer). The change observed in detergent solubilization and the subsequent increase in activity in both FP-TAMRA labelling and substrate cleavage that has been shown in previous publications on the matter are indicative of the dynamics of TM 5 -suspected to be a substrate gate- being more dynamic and more constitutively open in detergent micelles (i.e. a non-native membrane environment). This has been observed in various positions of the TM5 X-ray structures obtained in detergent micelles, and the dynamics directly explored with ssNMR in liposomes (10.1021/jacs.9b08952). Therefore, the native-like activity observed for

DIBMALP-GlpG compared to the detergent micelles is indicative of a stabilization due to the presence of lateral pressure (i.e. a membrane or polymer belt), at least when compared to the TM dynamics. This is similar for the autoprocesing tendency of VcROM.

Interestingly, the change in FP-TAMRA labelling in this work (SI Fig 3B) is indicative of the exogenous lipids washed into protease-containing DIBMALPs affecting this dynamics of this region even within the membrane.

Reviewer #5 (Remarks to the Author):

In my opinion the authors have made appropriate changes based on my previous review and the article is now acceptable for publication.

We thank reviewer 5 for their helpful comments in the previous round of the review process.